# FULLY FIRST-ORDER METHODS FOR CONTEXTUAL STOCHASTIC BILEVEL OPTIMIZATION

## ABSTRACT

Contextual stochastic bilevel optimization (CSBO) is a new paradigm for decision making under uncertainty that generalizes stochastic bilevel optimization (SBO) by integrating contextual information in the lower level optimization problem and thus offers a stronger modeling capability. Nevertheless, owing to its semi-infinite nature, CSBO is extremely challenging from a computational perspective, hindering its real-world applications. Indeed, many algorithms designed for SBO are not applicable to CSBO. In this paper, we devise a double-loop fully first-order algorithm for solving CSBO and prove that both sample and gradient complexities of the algorithm are $\widetilde{\mathcal{O}}(\epsilon^{-8})$. To tackle the increasing number of inner loop iterations, we further develop an accelerated version of our algorithm using the random truncated multilevel Monte Carlo technique. The accelerated algorithm enjoys the improved complexities of $\widetilde{\mathcal{O}}(\epsilon^{-6})$. Our algorithms are fully first-order in the sense that they do not rely on second-order information, and hence these complexities cannot be directly compared with those of Hessian-based methods. Numerical experiments on meta-learning with real datasets demonstrate the superiority of the proposed algorithms, especially the accelerated version, over existing Hessian-based method in terms of both speed and accuracy.

## 1 INTRODUCTION

Contextual stochastic bilevel optimization (CSBO) is a new bilevel optimization framework introduced in Hu et al. (2023b) that accommodates contextual information or personalization in the lower level and takes the form

$$\min_{x \in \mathbb{R}^{d_x}} F(x) := \mathbb{E}_{\eta,\xi}[f(x, y^\star(x;\xi); \eta, \xi)]$$

$$\text{where} \quad y^\star(x;\xi) := \arg\min_{y \in \mathbb{R}^{d_y}} \mathbb{E}_{\eta|\xi}[g(x,y;\eta,\xi)] \quad \forall \xi \in \Xi \subseteq \mathbb{R}^{d_\xi}, x \in \mathbb{R}^{d_x}, \tag{1}$$

where we assume that $g(x,\cdot;\xi,\eta)$ is strongly convex for any $x \in \mathbb{R}^{d_x}, \xi \in \Xi$ and $\eta$ so that the lower level minimizer $y^*(x;\xi)$ is unique. Further assumptions on the functions $f$ and $g$ are presented in the next section. The upper level expectation is with respect to the joint distribution of the two random vectors $\xi$ and $\eta$, while the lower level expectation is with respect to the conditional random vector $\eta|\xi$. The support $\Xi$ of $\xi$ can possibly be uncountably infinite. We do not assume knowledge of the distributions but only access to i.i.d. samples from the marginal distribution $\mathbb{P}_\xi$ and the conditional distribution $\mathbb{P}_{\eta|\xi}$.

CSBO subsumes stochastic bilevel optimization (SBO) (Ghadimi & Wang, 2018; Kwon et al., 2023a) as a special case. Compared with SBO, CSBO offers two distinctive modeling advantages: (i) the lower-level decision $y$ in CSBO can be coupled not only with upper-level decision $x$ but also with side information $\xi$; (ii) the number of lower-level decision makers in CSBO can be arbitrary as $\Xi$ is not necessarily a finite set. Besides SBO, CSBO also generalizes contextual stochastic optimization (Bertsimas & Kallus, 2020) and conditional stochastic optimization (Hu et al., 2020a;b). Consequently, CSBO serves as a versatile modeling paradigm with a wide range of applications, such as meta-learning (Rajeswaran et al., 2019), end-to-end learning (Rychener et al., 2023), personalized federated learning (Shamsian et al., 2021), hierarchical representation learning (Yao et al., 2019), Wasserstein DRO with side information (Yang et al., 2022; Donti et al., 2017), and instrumental variable regression (Muandet et al., 2020; Kwon et al., 2023a).

The modeling power comes at a cost though: CSBO is extremely challenging from a computational perspective. Indeed, many algorithms designed for SBO are inapplicable to CSBO. For instance, numerous single-loop methods that are efficient for SBO (Guo et al., 2021a;b; Chen et al., 2021; 2022b; 2023; Hong et al., 2023; Yang et al., 2021) cannot be directly applied to solve CSBO. The fundamental distinction arises from the nature of the lower-level optimal solution. In SBO, the lower-level solution $y^\star(x)$ is a function solely of the upper-level decision $x$, while in CSBO, the lower-level decision $y^\star(x; \xi)$ depends not only on $x$ but also on the random variable $\xi$ representing contextual information. This difficulty also invalidates the warm-start strategies used in many double-loop SBO algorithms (Kwon et al., 2023a;b; Chen et al., 2025b; Gong et al., 2024), degrading their practical performance as well as theoretical guarantees. Motivated by the gap in the algorithmic development between SBO and CSBO, a recent work (Bouscary et al., 2025) proposes a framework to reformulate CSBO as SBO, thereby solving CSBO via SBO algorithms. However, their framework requires the additional assumption that the lower-level objective function $\mathbb{E}_{\eta|\xi}[g(x, y; \eta, \xi)]$ is analytic with respect to $(y, \xi)$ and that $\xi$ is either a discrete random variable with a finite support (*i.e.*, $|\Xi| < \infty$) or a continuous random variable whose density function is uniformly bounded away from 0.

For general CSBO, Hu et al. (2023b) developed a double-loop algorithm. The outer loop adopts a vanilla stochastic gradient descent framework for the upper-level problem, and the inner loop is to compute the lower-level minimizer $y^\star(x; \xi)$ for constructing an upper-level gradient estimator. This algorithm achieves gradient and sample complexities of $\widetilde{\mathcal{O}}(\epsilon^{-6})$ (Hu et al., 2023b). To alleviate the large number of inner iterations, Hu et al. (2023b) further employed the random truncated multi-level Monte Carlo (RT-MLMC) technique to develop an accelerated algorithm that enjoys the strengthened complexities $\widetilde{\mathcal{O}}(\epsilon^{-4})$. A shared drawback of these two algorithms is that each iteration requires computing multiple Hessian estimators and a matrix of mixed second-order derivatives of $g$, which leads to high per-iteration computational costs and slow performance in practice.

In view of the above discussions, this paper aims to develop a fully first-order, Hessian-free algorithm for solving general CSBO problems. Our contributions are as follows.

- We propose a fully first-order double-loop algorithm (c.f. Algorithm 1) for CSBO and prove that its sample and gradient complexities are both $\widetilde{\mathcal{O}}(\epsilon^{-8})$. Our algorithmic framework differs fundamentally from that in (Hu et al., 2023b) and is based on a suitable penalty formulation of problem (1). To the best of our knowledge, this is the first fully first-order algorithm for solving general CSBO problems that does not rely on any second-order oracle of $g$.

- To circumvent the increasing number of inner iterations of our proposed double-loop algorithm, we devise an accelerated variant of our algorithm by invoking the RT-MLMC technique. We also show that this accelerated algorithm enjoys the improved sample and gradient complexities of $\widetilde{\mathcal{O}}(\epsilon^{-6})$.

- Unlike the situation in (Hu et al., 2023b), a straightforward adoption of their RT-MLMC technique in our algorithmic framework will introduce a large variance to the resulting algorithm, which significantly affects its practical performance. This is mainly due to the increasing penalty parameter in our algorithmic framework, which amplifies the variance of the gradient estimator. To cope with this issue, we develop a novel stepsize strategy for our accelerated algorithm that can effectively control the instability without compromising the theoretical complexity. To the best of our knowledge, this is the first adaptive stepsize strategy for controlling the overall variance in RT-MLMC-based gradient algorithms, which could be of independent interest.

- We demonstrate the superiority of our proposed algorithms, especially the accelerated one, over the Hessian-based algorithm in (Hu et al., 2023b) via numerical experiments on a meta-learning application with using the tinyImageNet datasets (Mnmoustafa, 2017).

Finally, we should point out that our complexities $\widetilde{\mathcal{O}}(\epsilon^{-8})$ and $\widetilde{\mathcal{O}}(\epsilon^{-6})$ for the basic and accelerated algorithms should not be directly compared with the corresponding ones in Hu et al. (2023b), as we do not rely on any second-order oracles and thus have much smaller per-iteration computational costs.

## 1.1 Related Work

Bilevel optimization is a big topic with a long history. Below we provide a brief review of its recent development, with a focus on SBO and CSBO. Assuming second-order oracles of $g$, Ghadimi & Wang (2018) developed the first SBO algorithm with a provable non-asymptotic complexity guarantee. In their approach, the inner loop computes an approximate solution for $y^\star(x)$, which is then used to estimate the gradient of $F$. This seminal work has spurred the development of a diverse suite of methods using second-order oracles; to name a few, stocBiO in Ji et al. (2020), SOBA and SABA in Dagréou et al. (2022), MDBO for distributed SBO in Gao et al. (2023), TTSA algorithm in Hong et al. (2023), SUSTIAN algorithm in Khanduri et al. (2021), ALSET method in Chen et al. (2021), SVRB in Guo et al. (2021a), BSVRB in Hu et al. (2023a). Note that SVRB and BSVRB apply to SBO with multiple lower-level problems, which is a special case of CSBO when the realization of $\xi$ is finite.

The heavy cost caused by the computation of second-order derivatives and inverse Hessian of $g$, required by aforementioned algorithms, motivates the exploration of fully first-order methods for solving SBO problems, pioneered by Kwon et al. (2023a). Many fully first-order algorithms have been developed subsequently for SBO. Within this fully first-order paradigm, the prevalent approaches mainly fall into two classes: (i) single-loop first-order methods, which adopt a Lagrangian- or penalty-type scheme, such as Kwon et al. (2023b); (ii) double-loop first-order methods, which maintain the bilevel hierarchy in the algorithmic design and often utilize a warm-start strategy for the lower-level optimization to enhance efficiency, such as F$^2$SA-$p$ in Chen et al. (2024), F$^2$BA and F$^2$BSA in Chen et al. (2025a).

A common feature shared by both the aforementioned single-loop and double-loop algorithms is the exploitation of the fact that the optimal solution $y^\star(x)$ to the lower-level problem in SBO depends only on $x$. In contrast, for CSBO, the lower-level minimizer $y^*(x;\xi)$ depends not only on $x$ but also on the side information variable $\xi$. This critical difference between CSBO and SBO hinders the direct application of these algorithms to general CSBO: algorithms for general CSBO cannot utilize information obtained from previous inner-loop iterations. This presents significantly greater analytical and computational challenges in CSBO than SBO.

In the context of CSBO, Hu et al. (2023b) devised a double-loop algorithm that relies on second-order oracles. Furthermore, the authors integrate the random truncated multilevel Monte Carlo (RT-MLMC) technique into their algorithmic framework to accelerate the proposed double-loop algorithm. Recently, Bouscary et al. (2025) provides an alternative approach for solving CSBO by reformulating it as a SBO problem to apply standard SBO algorithms. However, as pointed out previously, their approach requires the analyticity of the lower-level objective function and some assumption on the random variable $\xi$, which may limit its applicability. *i.e.,* when the lower-level problems are (contextual) RL problems. Leveraging the special structure of RL, their hypergradient formulation does not rely on second-order information. However, this observation does not apply to CSBO. Several papers study bilevel reinforcement learning Chen et al. (2022a); Chakraborty et al. (2024); Shen et al. (2025); Yang et al. (2025).

## 1.2 Preliminaries and Notation.

The symbol $\widetilde{\mathcal{O}}$ is a variant of the big-O notation that hides polylogarithmic factors. For an integer $M$, we let $[M] := \{1, \ldots, M\}$. Let $\psi : \mathbb{R}^{d_x} \times \mathbb{R}^{d_y} \to \mathbb{R}$ be a function, its gradients with respect to $(x, y)$, $x$ and $y$ are denoted by $\nabla\psi$, $\nabla_1\psi$, $\nabla_2\psi$, respectively. The Hessian of $\psi$ with respect to $(x, y)$, $x$ and $y$ are similarly denoted by $\nabla^2\psi$, $\nabla_{11}^2\psi$ and $\nabla_{22}^2\psi$, while $\nabla_{12}^2\psi$ and $\nabla_{21}^2\psi$ are $d_x \times d_y$ and $d_y \times d_x$ matrices whose $(i,j)$-th elements are $\partial^2_{x_iy_j}\psi$ and $\partial^2_{y_ix_j}\psi$, respectively. We say $\psi$ is $L$-Lipschitz continuous if for any $(x_1, y_1) \in \mathbb{R}^{d_x} \times \mathbb{R}^{d_y}$ and $(x_2, y_2) \in \mathbb{R}^{d_x} \times \mathbb{R}^{d_y}$, we have $\|\psi(x_1, y_1) - \psi(x_2, y_2)\| \le L\|(x_1, y_1) - (x_2, y_2)\|$. It is further called $S$-smooth if it is differentiable and its gradient is $S$-Lipschitz continuous. If $\psi - \frac{\mu}{2}\|\cdot\|^2$ is convex, then $\psi$ is said to be $\mu$-strongly convex. For a vector-valued function $h : \mathbb{R}^{d_a} \to \mathbb{R}^{d_b}$, the Jacobian matrix is defined to be the $d_b \times d_a$ matrix $Dh := [\nabla h_1, \cdots, \nabla h_{d_b}]^\top$. For $z = (z_1, z_2)$, the partial derivative of $h$ with respect to $z_1$ is denoted as $D_{z_1}h$. For sequences $\{x_k\}_k$, $\{y_k\}_k$, $\{z_k\}_k$ generated by Algorithm 1 or Algorithm 2, we denote the corresponding $\sigma$-algebra by $\mathcal{F}_k := \sigma\{x_0, y_0, z_0; x_1, y_1, z_1, \cdots, x_k, y_k, z_k\}$.

**Algorithm 1**

**Input:** $x_0 \in \mathbb{R}^{d_x}$.

1: **for** $k = 1, \cdots, K$ **do**
2:      Set $\lambda_k = \frac{2\ell_{f,1}}{\mu_g}(k+1)^{1/4}$, $\alpha_k = \frac{\mathcal{O}(1)}{\sqrt{k+1}}$, $T_k = k$.
3:      Sample $\xi_k$ from $\mathbb{P}_\xi$, set $y_k^0 = z_k^0$.
4:      **for** $t = 0, 1, \cdots, T_k - 1$ **do**
5:          Sample $\eta_k^t$ from $\mathbb{P}_{\eta|\xi_k}$.
6:          Set $\beta_t = \frac{8}{\mu_g(t+1)}$.
7:          $y_k^{t+1} = y_k^t - \beta_t \nabla_2 g(x_k, y_k^t; \eta_k^t, \xi_k)$
8:          $z_k^{t+1} = z_k^t - \frac{\beta_t}{\lambda_k} \nabla_z L(x_k, z_k^t, y_k^t, \lambda_k; \eta_k^t, \xi_k)$
9:      **end for**
10:     Set $z_{k+1} = z_k^{T_k}$, $y_{k+1} = y_k^{T_k}$
11:     Sample $\eta_k$ from $\mathbb{P}_{\eta|\xi_k}$
12:     $x_{k+1} = x_k - \alpha_k \nabla_x L(x_k, z_{k+1}, y_{k+1}, \lambda_k; \eta_k, \xi_k)$
13: **end for**

**Output:** $x_{K+1}$

## 2   ALGORITHMS

Our algorithms and theoretical analysis rely on the following assumptions. Similar assumptions also appear in the literature of SBO and CSBO Ghadimi & Wang (2018); Guo et al. (2021a); Chen et al. (2021; 2022b); Hong et al. (2023); Hu et al. (2023b).

**Assumption 2.1.** *Problem* (1) *satisfies the following regularity conditions:*

    *(i) For any $\eta$ and $\xi$, $f(x, y; \xi, \eta)$ is continuously differentiable and $g(x, y; \xi, \eta)$ is twice continuously differentiable in $x$ and $y$.*

    *(ii) For any $x$, $\eta$ and $\xi$, $g(x, y; \xi, \eta)$ is $\mu_g$-strongly convex in $y$.*

    *(iii) For any $\eta$ and $\xi$, $f(x, y; \xi, \eta), \nabla f(x, y; \xi, \eta)$, $\nabla g(x, y; \xi, \eta)$, and $\nabla^2 g(x, y; \xi, \eta)$ are $\ell_{f,0}$, $\ell_{f,1}$, $\ell_{g,1}$, and $\ell_{g,2}$-Lipschitz continuous in $(x, y)$, respectively.*

    *(iv) For any $x \in \mathbb{R}^{d_x}$ and $y \in \mathbb{R}^{d_y}$, there exist $\tau_f > 0$ and $\tau_g > 0$ such that*

$$\mathbb{E}[\|\nabla f(x, y; \eta, \xi) - \mathbb{E}[\nabla f(x, y; \eta, \xi) \mid \xi]\|^2 \mid \xi] \leq \tau_f^2,$$
$$\mathbb{E}[\|\nabla g(x, y; \eta, \xi) - \mathbb{E}[\nabla g(x, y; \eta, \xi) \mid \xi]\|^2 \mid \xi] \leq \tau_g^2.$$

Assumption 2.1(i)-(iii) imply in particular that for any $(x, y) \in \mathbb{R}^{d_x} \times \mathbb{R}^{d_y}$ and $(\xi, \eta) \sim \mathbb{P}_{\xi, \eta}$, $\nabla f(x, y; \eta, \xi)$ and $\nabla g(x, y; \eta, \xi)$ are unbiased estimators for $\nabla F(x, y; \eta, \xi)$ and $\nabla \mathbb{E}_{\eta|\xi}[g(x, y; \eta, \xi)]$, and that $F$ is $\ell_{F,1}$-smooth; see Lemma B.4.

### 2.1   THE BASIC ALGORITHM

We first present a basic algorithm for problem (1); see Algorithm 1. To begin, note that problem (1) is equivalent to the following problem:

$$\min_{x \in \mathbb{R}^{d_x}, z \in \mathbb{R}^{d_y}} \quad \mathbb{E}_{\eta, \xi}[f(x, z; \eta, \xi)]$$
$$\text{s.t.} \quad \mathbb{E}_{\eta|\xi}[g(x, z; \eta, \xi)] - \min_{y \in \mathbb{R}^{d_y}}[g(x, y; \eta, \xi)] \leq 0 \quad \forall \xi \in \Xi, x \in \mathbb{R}^{d_x}. \tag{2}$$

**Remark 2.2.** *The choice of $\mathcal{O}(1)$ in $\alpha_k = \frac{\mathcal{O}(1)}{\sqrt{k+1}}$ of Algorithm 1 and $\mathcal{O}(1)$ in $\alpha_0 = \mathcal{O}(1)\epsilon^4$ of Algorithm 2 are constant independent of $k$ and $\epsilon$.*

Inspired by Kwon et al. (2023a), our algorithms leverage the following penalty function:

$$L(x, z, y, \lambda; \eta, \xi) := f(x, z; \eta, \xi) + \lambda(g(x, z; \eta, \xi) - g(x, y; \eta, \xi)). \tag{3}$$

Very roughly speaking, the idea of our algorithm is to estimate $\nabla F$ using $\nabla_x L$, and then perform stochastic gradient descent. To do so, we denote $\bar{g}(x, y; \xi) := \mathbb{E}_{\eta|\xi}[g(x, y; \eta, \xi)]$, $\bar{f}(x, y; \xi) := \mathbb{E}_{\eta|\xi}[f(x, y; \eta, \xi)]$, and consider the following optimization problem with $\delta \in [0, 1)$:

$$\min_y \ Q(x, y, \delta; \xi) := \ \bar{g}(x, y; \xi) + \delta \bar{f}(x, y; \xi). \tag{4}$$

Denote its solution as $y^\star(x, \delta; \xi)$. Then, we can apply the chain rule to obtain

$$\begin{aligned}
\nabla F(x) &= \mathbb{E}_\xi[\nabla_1 \bar{f}(x, y^\star(x, 0; \xi); \xi) + D_x y^\star(x, 0; \xi)^\top \nabla_2 \bar{f}(x, y^\star(x, 0; \xi); \xi)] \\
&= \mathbb{E}_\xi[\nabla_1 \bar{f}(x, y^\star(x, 0; \xi); \xi) \\
&\quad - \nabla_{12}^2 \bar{g}(x, y^\star(x, 0; \xi); \xi)(\nabla_{22}^2 \bar{g}(x_k, y^\star(x, 0; \xi); \xi))^{-1} \nabla_2 \bar{f}(x, y^\star(x, 0; \xi); \xi)],
\end{aligned} \tag{5}$$

where the second equality follows from equality (15) in Appendix B.5. Notice that the right hand side of (5) involves gradients of $\bar{f}$ and Hessian of $\bar{g}$ at $x$ and $y^\star(x, 0; \xi)$. Nevertheless, we shall show in Lemma B.8 that this can be indeed approximated by $\nabla_x L(x, y^\star(x, \frac{1}{\lambda}; \xi), y^\star(x, 0; \xi), \lambda; \eta, \xi)$. Then, the inner loop of the $k$-th outer iteration (*i.e.*, steps 4-10 of Algorithm 1) executes a SGD-type algorithm to minimize $Q(x_k, y, 0; \xi_k)$ and $Q(x_k, y, \frac{1}{\lambda_k}; \xi_k)$. So, its outputs $y_{k+1}$ and $z_{k+1}$ approximate $y^*(x_k, 0; \xi)$ and $y^*(x_k, \frac{1}{\lambda_k}; \xi)$, respectively; see Lemma B.6. Therefore, we can estimate $\nabla F$ using only first-order information of $L$; see Appendix B.5 for a comprehensive discussion.

## 2.2 DERIVATION OF ALGORITHM 2

Noticing that in Algorithm 1, the number of inner iterations increases with the outer iteration counter $k$, which results in a heavy computational burden for large $k$. To tackle this, we develop an accelerated algorithm using the RT-MLMC technique Hu et al. (2023b; 2021); see Algorithm 2. For simplicity, we denote

$$u_k(t, \lambda) := \nabla_x L(x_k, z_k^{2^t}(\lambda), y_k^{2^t}, \lambda; \eta_k, \xi_k), \tag{6}$$

where the subscript $k$ denotes the iteration count of the outer loop, $t$ indicates the corresponding iteration count of inner loop is $2^t$, and $z_k^{2^t}(\lambda)$ and $y_k^{2^t}$ are inner iterates defined in steps 10 and 9 in Algorithm 2, respectively. It is a hypergradient estimator with $2^t$ inner iterations. To avoid the large number of inner iterations, we construct the gradient estimator for Algorithm 2 leveraging the following observation. By telescoping,

$$\begin{aligned}
&u_k(N, \lambda_N) \\
&= u_k(0, \lambda_0) + \sum_{n=1}^N p_n \frac{(u_k(n, \lambda_n) - u_k(n-1, \lambda_{n-1}))}{p_n} \\
&= u_k(0, \lambda_0) + \mathbb{E}_{\bar{n} \sim \mathbb{P}_N}\left[ \frac{u_k(\bar{n}, \lambda_{\bar{n}}) - u_k(\bar{n}-1, \lambda_{\bar{n}-1})}{p_{\bar{n}}} \right],
\end{aligned} \tag{7}$$

where $\mathbb{P}_N$ is the truncated geometric distribution with the upper bound $N$ and $\mathbb{P}_N(\bar{n} = n) = p_n \propto 2^{-n}$ for every $n \in [N]$. Equations (6) and (7) together suggest that one could replace the gradient estimator $\nabla_x L(x_k, z_{k+1}, y_{k+1}, \lambda_k; \eta_k, \xi_k)$ in Algorithm 1 with the following estimator.

$$u_k(0, \lambda_0) + p_{n_k}^{-1}(u_k(n_k, \lambda_{n_k}) - u_k(n_k - 1, \lambda_{n_k-1})), \tag{8}$$

where $n_k$ is a realization of the truncated geometric random variable with the upper bound $N$. Both gradient estimators admit the same bias but the estimator (8) has a much smaller computational cost on average via a proper selection of $\mathbb{P}_N$ that assigns a small probability to generate a large $\bar{n}$ and a large probability to generate a small $\bar{n}$.

Unlike Hu et al. (2023b), the integration of RT-MLMC technique into our algorithm is obstructed by additional challenges. More precisely, in our penalty-based algorithmic framework, in order for $\nabla_x L$ to be an accurate approximation of $\nabla F$, the penalty parameter $\lambda$ must grow sufficiently fast. Unfortunately, this will amplify the variance of the RT-MLMC gradient estimator. As a result, despite achieving accelerated complexities of $\widetilde{\mathcal{O}}(\epsilon^{-6})$, the numerical performance is highly unstable due to the large variance. To tackle this issue, we have developed a novel adaptive stepsize strategy; see steps 15-19 in Algorithm 2. Specifically, if $n_k$ exceeds a given threshold, we scale the stepsize

**Algorithm 2**

**Input:** $x_0 \in \mathbb{R}^{d_x}, N = \mathcal{O}(1)\log(\epsilon^{-1}), \alpha_0 = \mathcal{O}(1)\epsilon^4, c_0 \in (0,1], a_1 \in (0,1)$

1: **for** $k = 1, \cdots, K$ **do**
2:     Sample $n_k$ from the truncated geometric distribution $\mathbb{P}_N$.
3:     Sample $\xi_k$ from $\mathbb{P}_\xi$
4:     Set $p_{n_k} \propto 2^{-n_k}, \lambda_{n_k} = \frac{2\ell_{f,1}}{\mu_g}(2^{n_k})^{\frac{1}{4}}$.
5:     Set $y_k^0 = z_k^0(\lambda_{n_k}) = z_k^0(\lambda_{n_k-1}) = z_k^0(\lambda_0)$.
6:     **for** $t = 0, 1, \cdots, 2^{n_k} - 1$ **do**
7:         Set $\beta_t = \frac{8}{\mu_g(t+1)}$.
8:         Sample $\eta_k^t$ from $\mathbb{P}_{\eta|\xi_k}$.
9:         $y_k^{t+1} = y_k^t - \beta_t \nabla_2 g(x_k, y_k^t; \eta_k^t, \xi_k)$
10:       $z_k^{t+1}(\lambda_{n_k}) = z_k^t(\lambda_{n_k}) - \frac{\beta_t}{\lambda_{n_k}}\nabla_z L(x_k, z_k^t(\lambda_{n_k}), y_k^{t+1}, \lambda_{n_k}; \eta_k^t, \xi_k)$
11:       $z_k^{t+1}(\lambda_{n_k-1}) = z_k^t(\lambda_{n_k-1}) - \frac{\beta_t}{\lambda_{n_k-1}}\nabla_z L(x_k, z_k^t(\lambda_{n_k-1}), y_k^{t+1}, \lambda_{n_k-1}; \eta_k^t, \xi_k)$.
12:     **end for**
13:     Set $y_k^{2^{n_k-1}}, y_k^{2^{n_k}}, z_k^{2^{n_k-1}}(\lambda_{n_k-1}), z_k^{2^{n_k}}(\lambda_{n_k})$
14:     Sample $\eta_k$ from $\mathbb{P}_{\eta|\xi_k}$.
15:     **if** $n_k > c_0 N$ **then**
16:         $\alpha = a_1 \alpha_0$
17:     **else**
18:         $\alpha = \alpha_0$
19:     **end if**
20:     $x_{k+1} = x_k - \alpha(u_k(0, \lambda_0) + p_{n_k}^{-1}[u_k(n_k, \lambda_{n_k}) - u_k(n_k - 1, \lambda_{n_k-1})])$
21: **end for**

**Output:** $x_{K+1}$

---

by a factor $a_1 \in (0,1)$. This stepsize strategy is compatible with RT-MLMC technique in the sense that the resulting algorithm, Algorithm 2, similarly enjoys the improved complexities $\widetilde{\mathcal{O}}(\epsilon^{-6})$. To the best of our knowledge, this is the first time such a stepsize strategy has been utilized to control the overall variance in RT-MLMC-type gradient methods. An empirical comparison of our Algorithm 2 with and without the adaptive stepsize strategy appears in Figure 4 in Section 4, which demonstrates the instability without using the adaptive stepsize and the significant improvement using it.

Finally, for the purpose of the theoretical analysis, we assume that the initialization gap in lower-level problems is bounded. This assumption is also used implicitly in (Hu et al., 2023b); see for example the proof of Lemma 3 therein. Moreover, when the support $\Xi$ is finite, this holds trivially.

**Assumption 2.3.** *There exists $b > 0$ such that $\mathbb{E}_{\eta|\xi_k}[g(x_k, y_k^0; \eta, \xi_k) - g(x_k, y^\star(x_k, \xi_k); \eta, \xi_k)] \leq b$ for any $k \geq 1$.*

## 3 COMPLEXITY ANALYSIS

Due to the bilevel structure and potential non-convexity of $f$, the objective function $F$ is in general nonconvex in $x$. Thus, giving the SGD-nature of our algorithms, we aim to find $\{x_k\}_{k \in [K]}$ satisfying $\frac{1}{K}\sum_{k=1}^K \mathbb{E}[\|\nabla F(x_k)\|^2] \leq \epsilon^2$, which is a common stationarity measure in bilevel optimization.

Our first main theoretical result concerns the gradient and sample complexities of Algorithm 1. The proof can be found in Appendix B.7.

**Theorem 3.1.** *Suppose that Assumptions 2.1 and 2.3 hold. For the sequence $\{x_k\}_{k \in [K]}$ generated by Algorithm 1, to ensure $\frac{1}{K}\sum_{k=1}^K \mathbb{E}[\|\nabla F(x_k)\|^2] \leq \epsilon^2$, it suffices to set $K = \widetilde{\mathcal{O}}(\epsilon^{-4})$. Moreover, the sample complexity of $\xi$ and the gradient complexities of $\nabla_1 f, \nabla_1 g$ are of order $\widetilde{\mathcal{O}}(\epsilon^{-4})$, the sample complexity of $\eta$ and the gradient complexities of $\nabla_2 g, \nabla_2 f$ are of order $\widetilde{\mathcal{O}}(\epsilon^{-8})$.*

Thanks to the RT-MLMC technique, which greatly reduces the average number of inner iterations, we next show that the theoretical complexities are improved. Before presenting the theorem of

complexities, we first analyze the variance of the RT-MLMC gradient estimator in (8), summarized in the following Lemma, with more details appeared in Appendix B.6.

**Lemma 3.2.** *Under Assumptions 2.1 and 2.3, consider Algorithm 2, we have*

$$\mathbb{E}[\|\mathbb{E}[\nabla_x L(x_k, z_k^{2^N-1}(\lambda_N), y_k^{2^N-1}, \lambda_N; \eta_k, \xi_k) \mid \mathcal{F}_k]$$
$$- (u_k(0, \lambda_0) + p_{n_k}^{-1}(u_k(n_k, \lambda_{n_k}) - u_k(n_k - 1, \lambda_{n_k-1})))\|^2 \mid \mathcal{F}_k] \leq \mathcal{O}(2^{\frac{N}{2}}).$$

**Remark 3.3.** *The variance of the Hessian-based RT-MLMC gradient estimator in Hu et al. (2023b) is $\mathcal{O}(\log(\epsilon^{-1}))$ (c.f., page 16 therein). Unlike Hessian-based algorithms, our Algorithm 2 uses only first-order information. Consequently, the corresponding penalty parameter amplifies the variance and requires additional treatment in the technical analysis. Specifically, with $N = 4\log(\epsilon^{-1})$ as defined in Algorithm 2, the variance of our RT-MLMC gradient estimator is $\mathcal{O}(2^{\frac{N}{2}}) = \widetilde{\mathcal{O}}(\epsilon^{-2})$. This leads to the following $\widetilde{\mathcal{O}}(\epsilon^{-6})$ sample complexity of $\eta$ for the accelerated algorithm (Algorithm 2).*

**Theorem 3.4.** *Suppose that Assumptions 2.1 and 2.3 hold. For the sequence $\{x_k\}_{k\in[K]}$ generated by Algorithm 2, to ensure $\frac{1}{K}\sum_{k=1}^K \mathbb{E}[\|\nabla F(x_k)\|^2] \leq \epsilon^2$, it suffices to set $K = \widetilde{\mathcal{O}}(\epsilon^{-6})$, $N = \mathcal{O}(1)\log(\epsilon^{-1})$ and $\alpha_0 = \mathcal{O}(1)\epsilon^4$. Moreover, the sample complexities of $\xi$ and $\eta$, and the gradient complexities of $\nabla_1 f$, $\nabla_1 g$ $\nabla_2 g$, and $\nabla_2 f$ are of order $\widetilde{\mathcal{O}}(\epsilon^{-6})$.*

We defer the proof to Appendix B.8. Note that the sample and gradient complexities of Algorithm 1 are $\widetilde{\mathcal{O}}(\epsilon^{-8})$ by Theorem 3.1. In contrast, although Algorithm 2 needs a larger $K$ compared to Algorithm 1, eventually its sample and gradient complexities are $\widetilde{\mathcal{O}}(\epsilon^{-6})$. Although our complexity results seem significantly weaker than the Hessian-based method in Hu et al. (2023b) ($\widetilde{\mathcal{O}}(\epsilon^{-6})$ for standard version and $\widetilde{\mathcal{O}}(\epsilon^{-4})$ for RT-MLMC accelerated version), as fully first-order methods, our algorithms only involve gradient computation and arithmetic operations. Instead, Hessian-based methods require computation of second-order oracles, which, despite the efficient implementation of Hessian inverse estimation using Hessian estimators demonstrated in Algorithm 4 in Hu et al. (2023b), is still computationally expensive. For example, consider the meta-learning problem in Section 4 numerical experiments, we can see that the per-iteration flops cost of our Algorithm 1 and Algorithm 2 is $\mathcal{O}(T_k d_y^2 + d_x)$, while it is $\mathcal{O}(N d_y^2 + d_x d_y + T_k d_y)$ in Hu et al. (2023b). It remains an interesting and open question if one could better control the increasing penalty parameter such that the variance of the RT-MLMC gradient estimator, as demonstrated in Lemma 3.2, could reduce from $\mathcal{O}(\epsilon^{-2})$ to $\mathcal{O}(\log(\epsilon^{-1}))$, which would lead to improved $\mathcal{O}(\epsilon^{-4})$ complexity of the accelerated methods. However, for fully first-order method to get $\mathcal{O}(\epsilon^{-4})$, it might require additional assumptions on higher-order smoothness. Nevertheless, our experimental results confirm the significant computational advantage of our fully first-order methods over Hessian-based approaches.

## 4 NUMERICAL EXPERIMENTS

In this section, we evaluate the performance of our proposed first-order algorithms using two examples: the meta-learning problems (Finn et al., 2017; Rajeswaran et al., 2019) and the Wasserstein Distributionally Robust Optimization with Side Information (WDRO-SI) (Yang et al., 2022; Hu et al., 2023b), and compare our methods with the RT-MLMC Hessian-based method in Hu et al. (2023b) and the reduction strategies in Bouscary et al. (2025) with the reformulated SBO problem solved by stocBiO in Ji et al. (2020).

Algorithm 1, Algorithm 2, the RT-MLMC Hessian-based method in Hu et al. (2023b), and the reduction strategies in Bouscary et al. (2025) (from now on, we call it by "reduction + stocBiO" for simplicity and clarity), as well as all experiments, are implemented in Julia 1.12, and are performed on an Apple Macbook pro with M4 Pro (14 cores) and 48G memory.

### 4.1 META-LEARNING

We consider the meta-learning problem in which there is a distribution over tasks ($\xi \sim \mathbb{P}_\xi$), each task comes with its own training data and validation data $\eta_\xi \sim \mathbb{P}_{\eta|\xi}$, and the goal is to learn a shared meta-parameter so that, for each task, adapting from the meta-parameter using the training data yields low loss on the validation data.

Formally, we consider the following meta-learning problem, a special case of the CSBO problem:

$$\min_{x \in \mathbb{R}^{d_x}} \mathbb{E}_{\xi \sim \mathbb{P}_\xi} \mathbb{E}_{\eta_\xi^{\text{val}} \sim \mathbb{P}_{\eta|\xi}} [l_\xi(y^\star(x;\xi), \eta_\xi^{\text{val}})]$$

$$\text{where} \quad y^\star(x;\xi) = \arg\min_{y \in \mathbb{R}^{d_y}} \mathbb{E}_{\eta_\xi^{\text{tr}} \sim \mathbb{P}_{\eta|\xi}} [l_\xi(y, \eta_\xi^{\text{tr}}) + \frac{\gamma}{2}\|y-x\|^2] \quad \forall \xi \in [M], x \in \mathbb{R}^{d_x}, \tag{9}$$

where $\mathbb{P}_\xi$ is the distribution over all $M$ tasks; $\mathbb{P}_{\eta|\xi}$ is the distribution of data from the task $\xi$; $\eta_\xi^{\text{tr}}$ and $\eta_\xi^{\text{val}}$ are the training and validation datasets for the task $\xi$, respectively; $x$ is the meta-parameter shared within all tasks; $y^\star(x;\xi)$ is the optimal parameter learned from a regularized problem corresponding to task $\xi$; $l_\xi$ is a loss function, and $\gamma > 0$ is a regularization hyperparameter.

We follow the settings in Hu et al. (2023b): for every task $\xi \in [M]$, the loss function $l_\xi$ is a multi-class logistic loss using a linear classifier parameterized by $y_\xi$, the regularization hyperparameter $\gamma$ is set to be 2, and the dataset is features of images in tinyImageNet (Mnmoustafa, 2017) extracted by the pre-trained ResNet-18 network (He et al., 2016). Specifically, we pick 5 tasks from tinyImageNet, and randomly select 10 classes of images from the 10 classes of similar objects in each task, with every class containing 500 images. Each image is resized and preprocessed by the pre-trained ResNet-18 network to be a 512-dimensional vector. 90% of the images are taken as training data, while the rest of the images are regarded as validation data.

For more detailed parameter settings of this numerical experiment, please see Appendix B.9.

We evaluate the performance via three measurements: the estimated upper-level objective function value, the estimated stationarity and the validation prediction errors. To compute these measurements, we first run each algorithm itself to obtain the corresponding sequence $\{x_k\}$. For each sequence $\{x_k\}$, we partition it into 100 equally spaced grid points, at which we evaluate the performance measurements. This is for saving time and is enough for comparison. Specifically, for each selected $x_k$, for every $\xi \in [5]$, we estimate $y_{k+1}^\xi$ and $z_{k+1}^\xi$ via 100 iterations of the lower-level updates, i.e., steps 4-9, of Algorithm 1, where each sampling of $\eta_\xi^{\text{tr}}$ returns the whole training set. Then the upper-level objective function value is estimated by computing the sample average of $l_\xi(y_{k+1}^\xi, \eta_\xi^{\text{val}})$ over $\xi \in [5]$ and the whole validation set; the stationarity is similarly estimated using the sample average over stationarities.

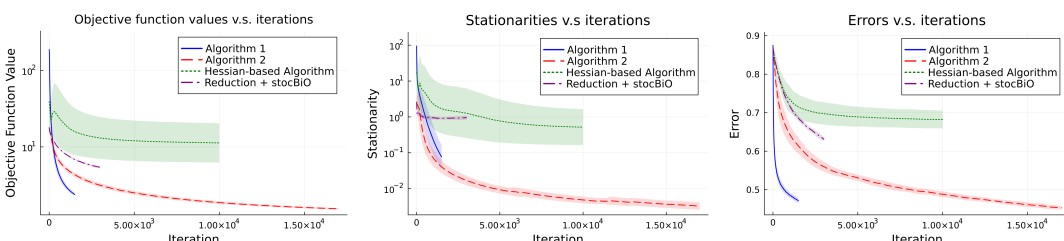

Figure 1: The measurements against outer iterations over meta-learning example. Error bars show $\pm 1$ standard deviation over 10 experiments. Note that the seemingly early stopping of Algorithm 1 is because Algorithm 1 runs so slow that exceeds the runtime range.

Figure 1 and Figure 2 show these measurements averaged over 10 experiments against the number of outer and inner iterations, respectively, while Figure 3 shows the averaged measurements against the computational time. Note that since we use a minibatch of $\xi$, the total number of inner iterations of two RT-MLMC methods are multiplied by 10. From the plots, Algorithm 1 exhibits the fastest decrease of objective function values and errors versus outer iteration in the first 1500 outer iterations, followed by Algorithm 2, then reduction + stocBiO, while the RT-MLMC Hessian-based method is the slowest one. However, when considered in terms of inner iterations and CPU computational time, Algorithm 2 achieves the greatest reduction of objective function values and stationarity, while the other three methods are overall comparable and are significantly slower than Algorithm 2. More importantly, despite the use of high basis degrees 50 for the reduction method, its stationarities remain remarkably higher throughout. For the prediction error, although Algorithm 2 initially lagged behind Algorithm 1, it ultimately surpassed Algorithm 1. Since the truncation level for RT-MLMC

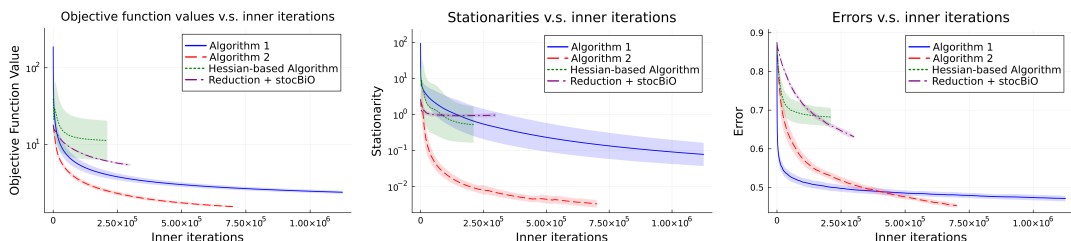

Figure 2: The measurements against inner iterations, each inner iteration refers to steps 4-9 of Algorithm 1, or steps 6-12 of Algorithm 2, or `EpochSGD` for RT-MLMC Hessian-based method in Hu et al. (2023b) or steps 5-6 in Algorithm 2 in Ji et al. (2020). Error bars show the standard deviation over 10 experiments. Note that the seemingly early stopping of the Hessian-based method is because it runs so slowly, due to the computation of second-order oracles, that it exceeds the runtime range.

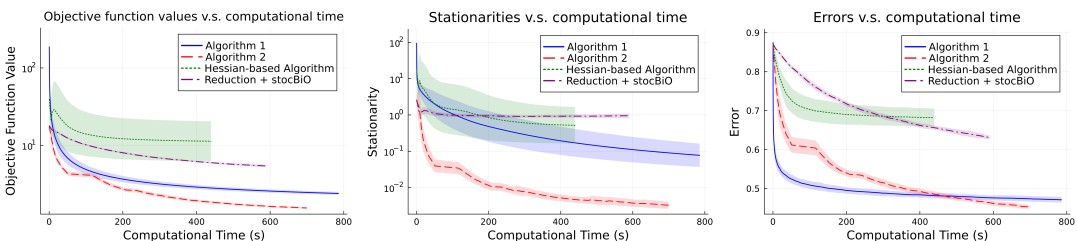

Figure 3: The measurements against computational time over meta-learning example. Error bars show standard deviation over 10 experiments.

Hessian-based method is $K = 12$, the number of inner iterations for RT-MLMC Hessian-based method is significantly lower than the other two methods, which still results in similar computational time, revealing that the heavy computational burden for Hessian-based method. Similarly, for reduction + stocBiO, since we use basis degrees 50, the dimension of lower-level problems is very high, leading to computational burden even heavier than RT-MLMC Hessian-based method. These behaviors confirm the advantages of our proposed fully first-order algorithms compared to RT-MLMC Hessian-based methods and reduction+stocBiO, and the efficiency of Algorithm 2 based on the RT-MLMC gradient estimation.

To demonstrate the effectiveness of our adaptive stepsize strategy, we conduct the same experiments using Algorithm 2 with and without the strategy by respectively setting $a_1 = 0.05$ and $a_1 = 1$, following the same settings described above. The results are presented in Figure 4. As shown, without the adaptive stepsize strategy, the results exhibit considerable variance (represented by the orange shaded area) and worse mean (the orange dash line), whereas with the adaptive stepsize strategy, the performance becomes substantially more stable. These results validate the practical usefulness of the adaptive stepsize strategy, which can empirically greatly reduce the variance of Algorithm 2 and the burden of tuning hyperparameter.

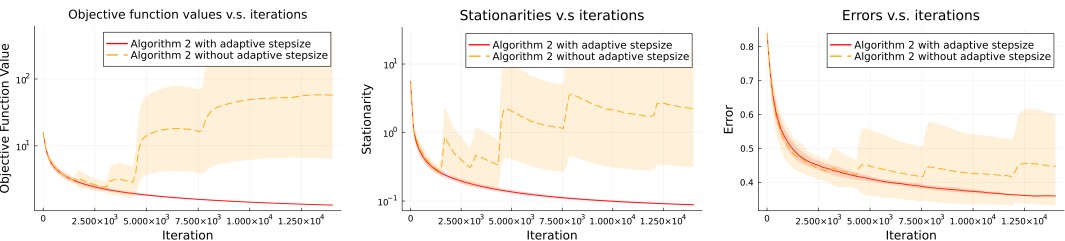

Figure 4: The comparison of Algorithm 2 with and without adaptive stepsize strategy over meta-learning example. Error bars show standard deviation over 10 experiments.

## 4.2 WASSERSTEIN DISTRIBUTIONALLY ROBUST OPTIMIZATION WITH SIDE INFORMATION

The Wasserstein Distributionally Robust Optimization with side information (WDRO-SI) (Yang et al., 2022) focuses on the problem of robust stochastic optimization with side information $\xi$ and dependent randomness $\eta$. It aims to learn a mapping $f$, parameterized by $x$, that maps $\xi$ to a decision $w$ which minimizes the expected loss $l(w; \eta)$, subject to robustness against worst-case deviations of the joint distribution $(\xi, \eta)$ from a nominal distribution $\mathbb{P}^0$. Using a dual reformulation, WDRO-SI can be cast as a contextual stochastic bilevel optimization (CSBO) problem Hu et al. (2023b):

$$\min_x \mathbb{E}_{\xi \sim \mathbb{P}_\xi^0} \mathbb{E}_{\eta \sim l_{\eta|\xi}^0} [l(f(x; y^\star(x;\xi), \eta)) - \gamma_1 \|y^\star(x;\xi) - \xi\|^2]$$
$$y^\star(x;\xi) := \arg\min_\delta \mathbb{E}_{\eta \sim \mathbb{P}_{\eta|\xi}^0} [-l(f(x;\delta), \eta) + \gamma_1 \|\delta - \xi\|^2], \quad \forall \delta, x. \tag{10}$$

where $l_\beta(w, \eta) := \frac{h}{\beta} \log(1 + e^{\beta(w-\eta)}) + \frac{b}{\beta} \log(1 + e^{\beta(w-\eta)})$ is the smoothed version of newsvendor loss function $l(w, \eta) := h(w - \eta)_+ + b(\eta - w)_+$ with $(\cdot)_+ = \max(\cdot, 0)$.

The results are shown in Figure 5 and Figure 6. We can see that our methods illustrate a good performance compared to Hessian-based methods and reduction+stocBiO. Note that since we do not use minibatch for Hessian-based method, it is very sensitive to stepsizes. To make sure it will not produce NaN, we need to set a very small stepsize, which leads to a super slow convergence, as shown in the plots.

For more detailed parameter settings of this numerical experiment, please see Appendix B.10.

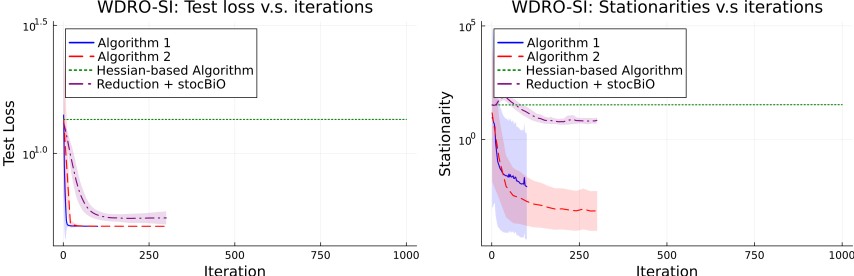

Figure 5: Test loss/stationarity again iterations over WDRO-SI example. Error bars show standard deviation over 10 experiments.

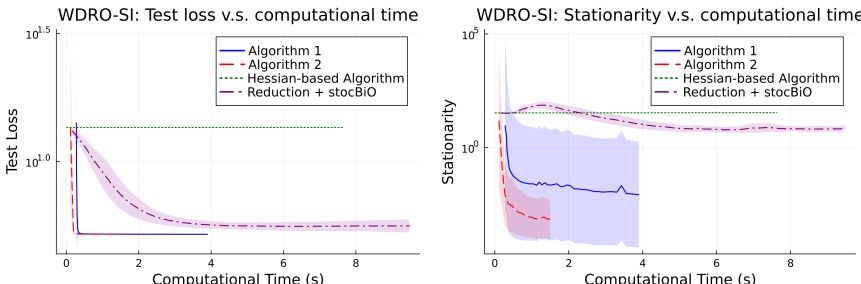

Figure 6: Test loss/stationarity again computational time over WDRO-SI example. Error bars show standard deviation over 10 experiments.

## REPRODUCIBILITY STATEMENT

All theoretical claims in this paper are accompanied by full proofs, which are included in the Appendix, and are cited explicitly from the main text. The numerical experiments are fully reproducible: we provide the complete implementation (Julia code), all scripts for data preprocessing, training, and evaluation, as part of the supplementary materials. Any parameters, random seeds, hardware details, and dependencies used are documented in the supplementary material.

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

# A  USE OF LARGE LANGUAGE MODELS (LLMS)

We used LLMs during the preparation of this manuscript in limited, well-defined ways, described below.

- We built the structure of the paper and wrote the core paragraphs ourselves. After that, we used LLMs to polish language, improve grammar, and enhance clarity and readability.
- In conducting the literature review, we used LLMs to help identify relevant papers we may originally have overlooked, to ensure thorough coverage.
- **No** theoretical results, proofs, algorithmic design, or experimental code were produced using LLMs; all substantive scientific contributions are our own.

We verified all content suggested by the LLMs. Any suggestions or drafts were carefully reviewed, edited, and corrected by us. We assume full responsibility for all content in this manuscript, including parts that were edited or polished via LLMs.

# B  PROOFS OF MAIN RESULTS

## B.1  METHODOLOGIES AND ROADMAP

The basic idea to construct a fully first-order algorithm for solving CSBO problems is to estimate $\nabla F$ using only first-order information of $f$ and $g$, and then perform stochastic gradient descent (SGD) for $F$. To do so, we first show in Lemma B.8 that $\nabla F$ can be approximated by $\mathbb{E}_{\eta,\xi}[\nabla_x L(x, y^\star(x, \frac{1}{\lambda}; \xi), y^\star(x, 0; \xi), \lambda; \eta, \xi)]$:

$$\mathbb{E}_{\eta,\xi}[\nabla_x L(x, y^\star(x, \frac{1}{\lambda}; \xi), y^\star(x, 0; \xi), \lambda; \eta, \xi)] \xrightarrow[\text{Lemma B.8}]{\text{Approximating}} \nabla F,$$

where $L$ is defined in (3), $y^\star(x, \delta; \xi)$ is the solution to (4). Then the inner loop of our algorithms is applying a SGD-type manner to minimize $Q(x_k, y, 0; \xi)$ and $Q(x_k, y, \frac{1}{\lambda_k}; \xi)$ for $y_{k+1}$ and $z_{k+1}$ that approximate $y^*(x_k, 0; \xi)$ and $y^*(x_k, \frac{1}{\lambda_k}; \xi)$, respectively:

$$\text{Sample } \xi_k \sim \mathbb{P}_\xi \to \left. \begin{array}{l} \min_y \ Q(x_k, y, 0; \xi_k) \xrightarrow{\text{SGD}} y_{k+1} \\[2mm] \min_y \ Q(x_k, y, \frac{1}{\lambda_k}; \xi_k) \xrightarrow{\text{SGD}} z_{k+1} \end{array} \right\} \xrightarrow[\text{Lemmas B.6 and B.9}]{\text{Approximating}} \begin{array}{l} \mathbb{E}_{\eta,\xi}[\nabla_x L(x, y^\star(x, \frac{1}{\lambda}; \xi), \\[2mm] y^\star(x, 0; \xi), \lambda; \eta, \xi)] \end{array}$$

Therefore, we use only the first-order information of $f$ and $g$, and only SGD-type methods to approximate $\nabla F$.

To further accelerate our Algorithm 1, we employ the multilevel Monte Carlo techniques, which, in addition to the previous framework, use extra approximations:

$$\text{Sample } \xi_k \sim \mathbb{P}_\xi, \text{ Sample } n_k \sim \mathbb{P}_N \to \left. \begin{array}{l} \min_y \ Q(x_k, y, 0; \xi_k) \xrightarrow{\text{SGD}} y_k^{2^{n_k - 1}}, y_k^{2^{n_k}} \\[2mm] \min_y \ Q(x_k, y, \frac{1}{\lambda_{n_k}}; \xi_k) \xrightarrow{\text{SGD}} z_k^{2^{n_k}}(\lambda_{n_k}) \\[2mm] \min_y \ Q(x_k, y, \frac{1}{\lambda_{n_k - 1}}; \xi_k) \xrightarrow{\text{SGD}} z_k^{2^{n_k - 1}}(\lambda_{n_k - 1}) \end{array} \right\}$$

$$\xrightarrow{\text{Obtain}} u_k(0, \lambda_0) + p_{n_k}^{-1}[u_k(n_k, \lambda_{n_k}) - u_k(n_k - 1, \lambda_{n_k - 1})] \xrightarrow[(7)]{\text{Approximating}} \begin{array}{l} \mathbb{E}_{\eta,\xi}[\nabla_x L(x_k, z_k^{2^N - 1}(\lambda_N), \\[2mm] y_k^{2^N - 1}, \lambda_N; \eta, \xi)] \end{array}$$

and

$$\mathbb{E}_{\eta,\xi}[\nabla_x L(x_k, z_k^{2^N - 1}(\lambda_N), y_k^{2^N - 1}, \lambda_N; \eta, \xi) \mid \mathcal{F}_k] \xrightarrow[\text{Lemma B.10}]{\text{Approximating}}$$

$$\mathbb{E}_{\eta,\xi}[\nabla_x L(x, y^\star(x, \frac{1}{\lambda}; \xi), y^\star(x, 0; \xi), \lambda; \eta, \xi)]$$

where $\mathbb{P}_N$ is the truncated geometric distribution whose upper bound is $N$ defined in Algorithm 2 and $p_k \propto 2^{-n_k}$; $\lambda_{n_k}, y_k^{2^{n_k - 1}}, y_k^{2^{n_k}}, z_k^{2^{n_k - 1}}(\lambda_{n_k - 1})$ and $z_k^{2^{n_k}}(\lambda_{n_k})$ are defined in Algorithm 2; $u_k$ is defined in (6).

## B.2 USEFUL LEMMA

**Lemma B.1.** *(Nesterov, 2018, Lemma 1.2.3) If $g : \mathbb{R}^d \to \mathbb{R}$ is continuously differentiable on $\mathbb{R}^d$. The first derivative of $g$ is Lipschitz continuous on $\mathbb{R}^d$ with constant $\ell_{g,1}$, then*

$$|g(y) - g(x) - \langle \nabla g(x), y - x \rangle| \le \frac{\ell_{g,2}}{2} \|y - x\|^2.$$

**Lemma B.2.** *(Nesterov, 2018, Lemma 1.2.4) If $g : \mathbb{R}^d \to \mathbb{R}$ is twice continuously differentiable on $\mathbb{R}^d$. The second derivative of $g$ is Lipschitz continuous on $\mathbb{R}^d$ with constant $\ell_{g,2}$, then*

$$\|\nabla g(y) - \nabla g(x) - \nabla^2 g(x)(y - x)\| \le \frac{\ell_{g,2}}{2} \|y - x\|^2$$

$$|g(y) - g(x) - \langle \nabla g(x), y - x \rangle - \frac{1}{2} \langle \nabla^2 g(x), y - x \rangle| \le \frac{\ell_{g,2}}{6} \|y - x\|^3.$$

Similar to the proof of (Nesterov, 2018, Lemma 1.2.3), we have the following result:

**Lemma B.3.** *Suppose $G : \mathbb{R}^d \to \mathbb{R}^m$ is continuously differentiable, and $DG : \mathbb{R}^d \to \mathbb{R}^{m \times d}$ is Lipschitz continuous with modulus $L$ in the following sense:*

$$\|DG(x) - DG(y)\|_2 \le L\|x - y\| \quad \forall x, y \in \mathbb{R}^d.$$

*where $\|\cdot\|_2$ denotes the spectral norm of matrices. Then, for all $x, y \in \mathbb{R}^d$, it holds that*

$$\|G(x) - G(y) - DG(x)(y - x)\| \le \frac{L}{2} \|x - y\|^2 \quad \forall x, y \in \mathbb{R}^d.$$

## B.3 THE SMOOTHNESS OF $F$

**Lemma B.4.** *(Xiao et al., 2023, Lemma 14) Under Assumption 2.1, there exists $\ell_{F,1} > 0$ such that*

$$\|\nabla F(x) - \nabla F(x')\| \leq \ell_{F,1}\|x - x'\|.$$

## B.4 THE CONVERGENCE RATE OF INNER LOOP

The next lemma shows that $Q(x, y, \delta; \xi)$ is strongly convex provided $\delta$ is sufficiently small. This is useful to analyze the convergence rate of the inner loop.

**Lemma B.5.** *Under Assumption 2.1, if $\delta < \frac{\mu_g}{\ell_{f,1}}$, then for any $\xi \in \Xi$, $Q(x, y, \delta; \xi)$ in (4) is $(\mu_g - \delta\ell_{f,1})$-strongly convex in $y$.*

*Proof.* It follows from Assumption 2.1(iii) that

$$\bar{f}(x, z_2; \xi) - \bar{f}(x, z_1; \xi) \leq \langle \nabla_2 \bar{f}(x, z_2; \xi), z_2 - z_1 \rangle + \frac{\ell_{f,1}}{2}\|z_1 - z_2\|^2.$$

Since $\bar{g}(x, y; \xi)$ is $\mu_g$-strongly convex in $y$, we have

$$\bar{g}(x, z_1; \xi) - \bar{g}(x, z_2; \xi) \geq \langle \nabla_2 \bar{g}(x, z_2; \xi), z_1 - z_2 \rangle + \frac{\mu_g}{2}\|z_1 - z_2\|^2.$$

Combining the above two inequalities, we get

$$\delta\bar{f}(x, z_1; \xi) - \delta\bar{f}(x, z_2; \xi) + \bar{g}(x, z_1; \xi) - \bar{g}(x, z_2; \xi)$$

$$\geq \delta\langle \nabla_2 \bar{f}(x, z_2; \xi), z_1 - z_2 \rangle - \frac{\delta\ell_{f,1}}{2}\|z_1 - z_2\|^2 + \langle \nabla_2 \bar{g}(x, z_2; \xi), z_1 - z_2 \rangle + \frac{\mu_g}{2}\|z_1 - z_2\|^2$$

$$= \langle \nabla_2 Q(x, z_2, \delta; \xi), z_1 - z_2 \rangle + (\frac{\mu_g}{2} - \frac{\delta\ell_{f,1}}{2})\|z_1 - z_2\|^2.$$

This completes the proof. □

The next lemma shows that the inner loop of Algorithm 1 and Algorithm 2 converges to $\left(y^\star(x_k, 0; \xi), y^\star(x_k, \frac{1}{\lambda_k}; \xi)\right)$ at a sublinear rate.

**Lemma B.6.** *Suppose that Assumptions 2.1 and 2.3 hold. Consider the $k$-th outer iteration of Algorithm 1 or Algorithm 2 with $x_k$ and $\lambda_k > \frac{\ell_{f,1}}{\mu_g}$. Then for $\{y_k^t\}_t$, $\{z_k^t\}_t$ generated by the inner loop of Algorithm 1 or Algorithm 2, we have*

$$\mathbb{E}[\|y_k^t - y^\star(x_k, 0; \xi)\|^2 \mid \mathcal{F}_k] \leq \mathcal{O}(\frac{1}{t}) \quad \text{and} \quad \mathbb{E}[\|z_k^t - y^\star(x_k, \frac{1}{\lambda_k}; \xi)\|^2 \mid \mathcal{F}_k] \leq \mathcal{O}(\frac{1}{t}).$$

*Proof.* It follows from the definition of $z_k^{t+1}$ in Algorithm 1 that

$$\|z_k^{t+1} - y^\star(x_k, \frac{1}{\lambda_k}; \xi_k)\|^2$$

$$= \|z_k^t - y^\star(x_k, \frac{1}{\lambda_k}; \xi_k)\|^2 + 2\langle z_k^{t+1} - z_k^t, z_k^t - y^\star(x_k, \frac{1}{\lambda_k}; \xi_k)\rangle + \|z_k^{t+1} - z_k^t\|^2$$

$$= \|z_k^t - y^\star(x_k, \frac{1}{\lambda_k}, \xi_k)\|^2 - 2\beta_t\langle \frac{1}{\lambda_k}\nabla_z L(x_k, z_k^t, y_k^t, \lambda_k; \eta_k^t, \xi_k), z_k^t - y^\star(x_k, \frac{1}{\lambda_k}; \xi_k)\rangle \qquad (11)$$

$$\quad + \|z_k^{t+1} - z_k^t\|^2$$

$$\leq -2\frac{\beta_t}{\lambda_k}(L(x_k, z_k^t, y_k^t, \lambda_k; \eta_k^t, \xi_k) - L(x_k, y^\star(x_k, \frac{1}{\lambda_k}; \xi_k), y_k^t, \lambda_k; \eta_k^t, \xi_k))$$

$$\quad + \|z_k^t - y^\star(x_k, \frac{1}{\lambda_k}, \xi_k)\|^2 + \|z_k^{t+1} - z_k^t\|^2,$$

where the inequality follows from the fact that $\frac{1}{\lambda_k}L(x_k, \cdot, y_k^t, \lambda; \eta_k^t, \xi_k)$ is $\mu_g - 1/\lambda_k\ell_{f,1}$-strongly convex (The strong convexity of this function can be established by a proof similar to that of

Lemma B.5). We now estimate the last term of the above inequality. Appealing again to the definition of $z_k^{t+1}$, we see that

$$\mathbb{E}_{\eta|\xi_k}[\|z_k^{t+1} - z_k^t\|^2]$$

$$= \mathbb{E}_{\eta|\xi_k}[\|\frac{\beta_t}{\lambda_k}\nabla_z L(x_k, z_k^t, y_k^t, \lambda_k; \eta_k^t, \xi_k)\|^2]$$

$$\leq 2\beta_t^2 \mathbb{E}_{\eta|\xi_k}\|\frac{1}{\lambda_k}\nabla_z L(x_k, z_k^t, y_k^t, \lambda_k; \eta_k^t, \xi_k) - \frac{1}{\lambda_k}\mathbb{E}_{\eta|\xi_k}[\nabla_x L(x_k, z_k^t, y_k^t, \lambda_k; \eta, \xi_k)]\|^2$$

$$+ 2\beta_t^2 \|\frac{1}{\lambda_k}\mathbb{E}_{\eta|\xi_k}[\nabla_x L(x_k, z_k^t, y_k^t, \lambda_k; \eta, \xi_k) - \frac{1}{\lambda_k}\mathbb{E}_{\eta|\xi_k}[\nabla_z L(x_k, y^\star(x_k, \frac{1}{\lambda_k}; \xi), y_k^t, \lambda_k; \eta, \xi_k)]\|^2$$

$$\leq 4\beta_t^2(\frac{\tau_f^2}{\lambda_k^2} + \tau_g^2) + 2\beta_t^2(\frac{4}{\lambda_k^2}\ell_{f,0}^2 + 2\ell_{g,1}^2 \mathbb{E}_{\eta|\xi_k}[\|z_k^t - y^\star(x_k, \frac{1}{\lambda_k}; \xi_k)\|^2])$$

(12)

where the first inequality follows from $\mathbb{E}_{\eta|\xi_k}[\nabla_z L(x_k, y^\star(x_k, \frac{1}{\lambda_k}; \xi), y_k^t; \eta, \xi_k)] = 0$ and the triangle inequality, the last inequality follows from Assumption 2.1(iv), the triangle inequality, $\|\bar{\nabla}_2 f(x, y; \xi)\| \leq \ell_{f,0}$ and the $\ell_{g,1}$-smoothness of $g$. Thus we have

$$\mathbb{E}_{\eta|\xi_k}[\|z_k^{t+1} - y^\star(x_k, \frac{1}{\lambda_k}; \xi_k)\|^2]$$

$$\leq (1 - \beta_t(\mu_g - \frac{\ell_{f,1}}{\lambda_k}))\mathbb{E}_{\eta|\xi_k}[\|z_k^t - y^\star(x_k, \frac{1}{\lambda_k}; \xi)\|^2]$$

$$+ 4\beta_t^2(\frac{\tau_f^2}{\lambda_k^2} + \tau_g^2) + 2\beta_t^2(\frac{4}{\lambda_k^2}\ell_{f,0}^2 + 2\ell_{g,1}^2 \mathbb{E}_{\eta|\xi_k}[\|z_k^t - y^\star(x_k, \frac{1}{\lambda_k}; \xi)\|^2])$$

(13)

$$\leq (1 - \frac{\beta_t\mu_g}{2} + 4\beta_t^2\ell_{g,1}^2)\mathbb{E}_{\eta|\xi_k}[\|z_k^t - y^\star(x_k, \frac{1}{\lambda_k}; \xi)\|^2] + \mathcal{O}(\frac{1}{t^2})$$

$$\leq (1 - \frac{\beta_t\mu_g}{4})\mathbb{E}_{\eta|\xi_k}[\|z_k^t - y^\star(x_k, \frac{1}{\lambda_k}; \xi)\|^2] + \mathcal{O}(\frac{1}{(t+1)^2}),$$

where the first inequality follows from the fact that $\frac{1}{\lambda_k}\mathbb{E}_{\eta|\xi_k}[L(x_k, z, y_k^t, \lambda_k; \eta, \xi_k)]$ is $(\mu_g - \ell_{f,1}/\lambda_k$-strongly convex in $z$, the second inequality follows by $\ell_{f,1}/\lambda_k \leq \mu_g/2$, the last inequality follows from $\beta_t \leq \mu_g/(16\ell_{g,1}^2)$. If $t \geq 1$, using $\beta_t = 8/(\mu_g(t+1))$ in Algorithm 1, multiplying both sides of the above inequality by $t(t+1)$ simultaneously will give the following inequality,

$$t(t+1)\mathbb{E}_{\eta|\xi_k}[\|z_k^{t+1} - y^\star(x_k, \frac{1}{\lambda_k}; \xi_k)\|^2]$$

$$\leq t(t-1)\mathbb{E}_{\eta|\xi_k}[\|z_k^t - y^\star(x_k, \frac{1}{\lambda_k}; \xi_k)\|^2] + \mathcal{O}(1)$$

(14)

$$\leq 2\mathbb{E}_{\eta|\xi_k}[\|z_k^0 - y^\star(x_k, \frac{1}{\lambda_k}; \xi_k)\|^2] + t\mathcal{O}(1),$$

where the second inequality is derived from the repeated use of the first inequality. Taking the expectation on both sides of the above inequality, we obtain

$$\mathbb{E}[\|z_k^{t+1} - y^\star(x_k, \frac{1}{\lambda_k}; \xi)\|^2 \mid \mathcal{F}_k] \leq \frac{2}{t(t+1)}\mathbb{E}[\|z_k^0 - y^\star(x_k, \frac{1}{\lambda_k}; \xi)\|^2 \mid \mathcal{F}_k] + \frac{\mathcal{O}(1)}{t}.$$

By a similar argument, we can obtain the convergence rate of $\{y_k^t\}_t$ that is generated by Algorithm 1, $\{y_k^t\}_t$, $\{z_k^t(\lambda_{n_k})\}_t$, $\{z_k^t(\lambda_{n_k-1})\}_t$ that are generated by Algorithm 2. This completes the proof.

□

## B.5 ESTIMATE BIAS

In this subsection, we shall show the bias of the gradient estimator of $\nabla F$ used in Algorithm 1 and Algorithm 2 is controllable. Specifically, we will show that $\|\nabla F(x_k) - \mathbb{E}[\nabla_x L(x_k, z_{k+1}, y_{k+1}, \lambda_k; \eta_k, \xi_k) \mid \mathcal{F}_k]\|$ and $\|\nabla F(x_k) - \mathbb{E}_{\eta,\xi,n_k}[(u_k(0, \lambda_0) + p_{n_k}^{-1}(u_k(n_k, \lambda_{n_k}) - u_k(n_k - 1, \lambda_{n_k-1}))) \mid \mathcal{F}_k]\|$ are upper bounded.

When $\delta$ in (4) is chosen such that $\delta < \frac{\mu_g}{\ell_{f,1}}$, by Lemma B.5, we know $Q(x, \cdot, \delta, \xi)$ is strongly convex, and hence the solution and the corresponding multiplier of (4) exist and are unique. The next Lemma shows that for any $\xi \in \Xi$, the solution to (4) $y^\star(x, \delta; \xi)$ is Lipschitz continuous in $\delta$ and $x$, respectively, provided $\delta$ is carefully selected.

**Lemma B.7.** *Under Assumption 2.1, if $0 \leq \delta' \leq \delta \leq \frac{\mu_g}{2\ell_{f,1}}$, there exist $\ell_{y,0}$, $\ell_{y,1}$, $\ell_{\mu,1}$ such that for any $\xi \in \Xi$*

$$\|y^\star(x, \delta; \xi) - y^\star(x, \delta'; \xi)\| \leq \ell_{y,0}|\delta - \delta'|,$$
$$\|y^\star(x, 0; \xi) - y^\star(x', 0; \xi)\| \leq \ell_{y,0}\|x - x'\|,$$

*where $\ell_{y,0} = \max\{\frac{\ell_{g,1} + \delta\ell_{f,1}}{\mu_g - \delta\ell_{f,1}}, \frac{\ell_{f,0}}{\mu_g - \delta\ell_{f,1}}\}$.*

*Proof.* By the definition of $y^\star(x_k, \delta; \xi)$ and the first-order necessary condition, we know that

$$\nabla_2\bar{g}(x_k, y^\star(x_k, \delta; \xi); \xi) + \delta\nabla_2\bar{f}(x_k, y^\star(x_k, \delta; \xi); \xi) = 0,$$

We take the derivative of both sides with respect to $x$ and $\delta$. Then, an application of the chain rule gives:

$$\nabla^2_{21}Q(x_k, y^\star(x_k, \delta; \xi)), \delta; \xi) + \nabla^2_{22}Q(x_k, y^\star(x_k, \delta; \xi), \delta; \xi)D_x y^\star(x_k, \delta; \xi) = 0,$$
$$\nabla_2\bar{f}(x_k, y^\star(x_k, \delta; \xi); \xi) + \nabla^2_{22}Q(x_k, y^\star(x_k, \delta; \xi), \delta; \xi)D_\delta y^\star(x_k, \delta; \xi) = 0.$$

where $Q(x, y, \delta; \xi)$ is defined in (4). By Lemma B.5 and the above two equalities, we have

$$D_x y^\star(x_k, \delta; \xi) = -(\nabla^2_{22}Q(x_k, y^\star(x_k, \delta; \xi)), \delta; \xi))^{-1}\nabla^2_{21}Q(x_k, y^\star(x_k, \delta; \xi), \delta; \xi),$$
$$D_\delta y^\star(x_k, \delta; \xi) = -(\nabla^2_{22}Q(x_k, y^\star(x_k, \delta; \xi), \delta; \xi))^{-1}\nabla_2\bar{f}(x_k, y^\star(x_k, \delta; \xi); \xi),$$

(15)

which imply

$$\|D_x y^\star(x_k, \delta; \xi)\| \leq \frac{\ell_{g,1} + \delta\ell_{f,1}}{\mu_g - \delta\ell_{f,1}} \quad \|D_\delta y^\star(x_k, \delta; \xi)\| \leq \frac{\ell_{f,0}}{\mu_g - \delta\ell_{f,1}}.$$

This completes the proof. $\qquad\square$

The following Lemma shows that $\nabla F(x)$ can be approximated using only first-order information of $L$, which plays a crucial role in our analysis.

**Lemma B.8.** *Suppose that Assumption 2.1 holds, and $\lambda > \frac{\ell_{f,1}}{\mu_g}$. Let the solution to (4) be $y^\star(x, \delta; \xi)$. Then we have*

$$\|\nabla F(x) - \mathbb{E}_{\eta,\xi}[\nabla_x L(x, y^\star(x, \frac{1}{\lambda}; \xi), y^\star(x, 0; \xi), \lambda; \eta, \xi)]\| = \mathcal{O}(\frac{1}{\lambda}).$$

*Proof.* By (15), we know that

$$D_x y^\star(x, 0; \xi) = -\nabla^2_{22}\bar{g}(x, y^\star(x, 0; \xi); \xi)^{-1}\nabla^2_{21}\bar{g}(x, y^\star(x, 0; \xi); \xi)$$

The above equality and the chain rule imply

$$\nabla F(x) = \mathbb{E}_\xi[\nabla_1\bar{f}(x, y^\star(x, 0; \xi); \xi) + D_x y^\star(x, 0; \xi)^\top \nabla_2\bar{f}(x, y^\star(x, 0; \xi); \xi)]$$
$$= \mathbb{E}_\xi[\nabla_1\bar{f}(x, y^\star(x, 0; \xi); \xi)$$
$$- \nabla^2_{12}\bar{g}(x, y^\star(x, 0; \xi); \xi)(\nabla^2_{22}\bar{g}(x_k, y^\star(x, 0; \xi); \xi))^{-1}\nabla_2\bar{f}(x, y^\star(x, 0; \xi); \xi)],$$

(16)

It follows from the definition of $L$ that

$$\mathbb{E}_{\eta,\xi}[\nabla_x L(x, y^\star(x, \frac{1}{\lambda}; \xi), y^\star(x, 0; \xi), \lambda; \eta, \xi)]$$
$$= \mathbb{E}_\xi\left[\nabla_1\bar{f}(x, y^\star(x, \frac{1}{\lambda}; \xi); \xi) + \lambda\left(\nabla_1\bar{g}(x, y^\star(x, \frac{1}{\lambda}; \xi); \xi); \xi) - \nabla_1\bar{g}(x, y^\star(x, 0; \xi); \xi)\right)\right]$$

(17)

By Lemma B.3,we know that

$$\nabla_1 \bar{g}(x, y^\star(x, \frac{1}{\lambda}; \xi); \xi) - \nabla_1 \bar{g}(x, y^\star(x, 0; \xi); \xi)$$

$$= \nabla_{12}^2 \bar{g}(x, y^\star(x, 0; \xi); \xi)(y^\star(x, \frac{1}{\lambda}; \xi) - y^\star(x, 0; \xi)) + r_1^g,$$

where $\|r_1^g\| = \mathcal{O}(1/\lambda^2)$. By Lemma B.3 and (15), we obtain

$$y^\star(x, \frac{1}{\lambda}; \xi) - y^\star(x, 0; \xi) = D_\delta y^\star(x, 0; \xi)(\frac{1}{\lambda} - 0) + r_2^g,$$

where $\|r_2^g\| = \mathcal{O}(1/\lambda^2)$. Using the expression for $D_\delta y^\star(x, 0; \xi)$ in (15), and combining the above equalities, we obtain

$$\nabla_1 \bar{g}(x, y^\star(x, \frac{1}{\lambda}; \xi); \xi) - \nabla_1 \bar{g}(x, y^\star(x, 0; \xi); \xi)$$

$$= \frac{1}{\lambda} \nabla_{12}^2 \bar{g}(x, y^\star(x, 0; \xi); \xi) \nabla_{22}^2 \bar{g}(x, y^\star(x, 0; \xi))^{-1} (\nabla_2 \bar{f}(x, y^\star(x, 0; \xi); \xi); \xi)) + r_3^g,$$

where $\|r_3^g\| = \mathcal{O}(1/\lambda^2)$. It follows from (16), (17) and the above equality that

$$\nabla F(x) - \mathbb{E}_{\eta, \xi}[\nabla_x L(x, y^\star(x, \frac{1}{\lambda}; \xi), y^\star(x, 0; \xi), \lambda; \eta, \xi)]$$

$$= \mathbb{E}_\xi[\nabla_1 \bar{f}(x, y^\star(x, 0; \xi); \xi) - \nabla_1 \bar{f}(x, y^\star(x, \frac{1}{\lambda}; \xi); \xi)] + r_4^g \tag{18}$$

where $\|r_4^g\| = \mathcal{O}(1/\lambda)$. Combining the Lipschitz property of $\bar{f}$, Lemma B.7 with the above equality yields this conclusion. $\square$

We now show a lemma stating that in the $k$-th outer iteration, we can use $\mathbb{E}[\nabla_x L(x_k, z_{k+1}, y_{k+1}, \lambda_k; \eta, \xi)]$ with $(y_{k+1}, z_{k+1})$ being obtained from the inner loop of Algorithm 1 to approximate $\mathbb{E}[\nabla_x L(x_k, y^\star(x_k, \frac{1}{\lambda_k}; \xi), y^\star(x_k, 0; \xi), \lambda_k; \eta, \xi)]$.

**Lemma B.9.** *Suppose that Assumptions 2.1 and 2.3 hold, consider Algorithm 1, we have*

$$\|\mathbb{E}[\nabla_x L(x_k, z_{k+1}, y_{k+1}, \lambda_k; \eta, \xi) - \nabla_x L(x_k, y^\star(x_k, \frac{1}{\lambda_k}; \xi), y^\star(x_k, 0; \xi), \lambda_k; \eta, \xi) \mid \mathcal{F}_k]\|^2 \leq \mathcal{O}(\frac{\lambda_k^2}{T_k}).$$

*Proof.* We have

$$\mathbb{E}[\nabla_x L(x_k, z_{k+1}, y_{k+1}, \lambda_k; \eta, \xi)] - \mathbb{E}[\nabla_x L(x_k, y^\star(x_k, \frac{1}{\lambda_k}; \xi), y^\star(x_k, 0; \xi), \lambda_k; \eta, \xi) \mid \mathcal{F}_k]$$

$$= \mathbb{E}[\nabla_1 \bar{f}(x_k, z_{k+1}, \xi) - \nabla_1 \bar{f}(x_k, y^\star(x_k, \frac{1}{\lambda_k}; \xi); \xi) \mid \mathcal{F}_k]$$

$$+ \lambda_k \mathbb{E}[\left( \nabla_1 \bar{g}(x_k, z_{k+1}; \xi) - \nabla_1 \bar{g}(x_k, y^\star(x_k, \frac{1}{\lambda_k}; \xi); \xi)) \right) \mid \mathcal{F}_k]$$

$$+ \lambda_k \mathbb{E}[(\nabla_1 \bar{g}(x_k, y^\star(x_k, 0; \xi); \xi) - \nabla_1 \bar{g}(x_k, y_{k+1}; \xi)) \mid \mathcal{F}_k], \tag{19}$$

which implies

$$\|\mathbb{E}[\nabla_x L(x_k, z_{k+1}, y_{k+1}, \lambda_k; \eta, \xi) - \nabla_x L(x_k, y^\star(x_k, \frac{1}{\lambda_k}; \xi), y^\star(x_k, 0; \xi), \lambda_k; \eta, \xi) \mid \mathcal{F}_k]\|^2$$

$$\leq \mathcal{O}(\lambda_k^2)(\mathbb{E}[\|z_{k+1} - y^\star(x_k, \frac{1}{\lambda_k}; \xi)\|^2 \mid \mathcal{F}_k] + \mathbb{E}[\|y_{k+1} - y^\star(x_k, 0; \xi)\|^2 \mid \mathcal{F}_k]) \leq \mathcal{O}(\frac{\lambda_k^2}{T_k}),$$

where the inequality follows from Lemma B.6. $\square$

Similar to the analysis in Lemma B.9, we can show the following result for Algorithm 2.

**Lemma B.10.** *Suppose that Assumptions 2.1 and 2.3 hold, consider Algorithm 2, we have*

$$\|\mathbb{E}[\nabla_x L(x_k, z_k^{2^N-1}(\lambda_N), y_k^{2^N-1}, \lambda_N; \eta, \xi) \mid \mathcal{F}_k]$$
$$- \mathbb{E}[\nabla_x L(x_k, y^\star(x_k, \frac{1}{\lambda_N}; \xi), y^\star(x_k, 0; \xi), \lambda_N; \eta, \xi) \mid \mathcal{F}_k]\|^2 \leq \mathcal{O}(\frac{\lambda_N^2}{2^N}).$$

Now, combining all lemmas in this subsection, we can upper bound $\|\nabla F(x_k) - \mathbb{E}[\nabla_x L(x_k, z_{k+1}, y_{k+1}, \lambda_k; \eta_k, \xi_k) \mid \mathcal{F}_k]\|$ and $\|\nabla F(x_k) - \mathbb{E}_{\eta, \xi, n_k}[(u_k(0, \lambda_0) + p_{n_k}^{-1}(u_k(n_k, \lambda_{n_k}) - u_k(n_k - 1, \lambda_{n_k-1}))) \mid \mathcal{F}_k]\|$ using triangle inequality. Then the bias of gradient estimator is controllable. The results are summarized in the following two lemmas.

**Lemma B.11.** *Under Assumptions 2.1 and 2.3, consider Algorithm 2, we have*

$$\|\nabla F(x_k) - \mathbb{E}[\nabla_x L(x_k, z_{k+1}, y_{k+1}, \lambda_k; \eta_k, \xi_k) \mid \mathcal{F}_k]\| \leq \mathcal{O}(\frac{1}{\lambda_k}) + \mathcal{O}(\frac{\lambda_k^2}{T_k}).$$

**Lemma B.12.** *Under Assumptions 2.1 and 2.3, consider Algorithm 2, we have*

$$\|\mathbb{E}[u_k(0, \lambda_0) + p_{n_k}^{-1}(u_k(n_k, \lambda_{n_k}) - u_k(n_k - 1, \lambda_{n_k-1})) \mid \mathcal{F}_k] - \nabla F(x_k)\|^2 \leq \mathcal{O}(\frac{1}{\lambda_N^2}).$$

*Proof.* By (7), we obtain

$$\mathbb{E}[u_k(0, \lambda_0) + p_{n_k}^{-1}(u_k(n_k, \lambda_{n_k}) - u_k(n_k - 1, \lambda_{n_k-1})) \mid \mathcal{F}_k]$$
$$= \mathbb{E}[\nabla_x L(x_k, z_k^{2^N-1}(\lambda_N), y_k^{2^N-1}, \lambda_N; \eta, \xi) \mid \mathcal{F}_k].$$

Then the desired result is due to Lemma B.8, Lemma B.10 and the above equality. $\square$

## B.6 THE VARIANCE OF RT-MLMC

Below, we demonstrate the variance of $u_k(0, \lambda_0) + p_{n_k}^{-1}(u_k(n_k, \lambda_{n_k}) - u_k(n_k - 1, \lambda_{n_k-1}))$ in Algorithm 2.

**Lemma B.13.** *Under Assumptions 2.1 and 2.3, consider Algorithm 2, we have*

$$\mathbb{E}[\|\mathbb{E}[\nabla_x L(x_k, z_k^{2^N-1}(\lambda_N), y_k^{2^N-1}, \lambda_N; \eta_k, \xi_k) \mid \mathcal{F}_k]$$
$$- (u_k(0, \lambda_0) + p_{n_k}^{-1}(u_k(n_k, \lambda_{n_k}) - u_k(n_k - 1, \lambda_{n_k-1})))\|^2 \mid \mathcal{F}_k] \leq \mathcal{O}(2^{\frac{N}{2}}).$$

*Proof.* It holds that

$$\mathbb{E}[\|\mathbb{E}[\nabla_x L(x_k, z_k^{2^N-1}(\lambda_N), y_k^{2^N-1}, \lambda_N; \eta_k, \xi_k) \mid \mathcal{F}_k]$$
$$- (u_k(0, \lambda_0) + p_{n_k}^{-1}(u_k(n_k, \lambda_{n_k}) - u_k(n_k - 1, \lambda_{n_k-1})))\|^2 \mid \mathcal{F}_k]$$
$$\leq 2\mathbb{E}[\|\mathbb{E}[\nabla_x L(x_k, z_k^{2^N-1}(\lambda_N), y_k^{2^N-1}, \lambda_N; \eta_k, \xi_k) \mid \mathcal{F}_k - u_k(0, \lambda_0)]\|^2 \mid \mathcal{F}_k]$$
$$+ 2\mathbb{E}[\|p_{n_k}^{-1}(u_k(n_k, \lambda_{n_k}) - u_k(n_k - 1, \lambda_{n_k-1}))\|^2 \mid \mathcal{F}_k].$$

Next, we analyze two terms on the right of the above inequality. For the first term, we have

$$\mathbb{E}[\|\mathbb{E}[\nabla_x L(x_k, z_k^{2^N-1}(\lambda_N), y_k^{2^N-1}, \lambda_N; \eta_k, \xi_k) \mid \mathcal{F}_k - u_k(0, \lambda_0)]\|^2 \mid \mathcal{F}_k]$$
$$\leq 6\ell_{f,0}^2 + 3\lambda_N^2 \ell_{g,1}^2 \mathbb{E}[\|z_k^{2^N-1}(\lambda_N) - y_k^{2^N-1}\|^2 \mid \mathcal{F}_k],$$

where the inequality follows from the definition $y_k^0$, $z_k^0(\lambda)$, $\mu_k^1$ and $\mu_k^2$, the smoothness of $f$ and Assumption 2.1(iii). Notice that

$$\mathbb{E}[\|z_k^{2^N-1}(\lambda_N) - y_k^{2^N-1}\|^2 \mid \mathcal{F}_k]$$
$$\leq \mathbb{E}[3\|z_k^{2^N-1}(\lambda_N) - y^\star(x_k, \frac{1}{\lambda_N}; \xi_k)\|^2 + 3\|y_k^{2^N-1} - y^\star(x_k, 0; \xi_k)\|^2 \mid \mathcal{F}_k]$$
$$+ 3\mathbb{E}[\|y^\star(x_k, \frac{1}{\lambda_N}; \xi_k) - y^\star(x_k, 0; \xi_k)\|^2 \mid \mathcal{F}_k]$$
$$\leq \mathcal{O}(\frac{1}{2^N - 1}) + \mathcal{O}(\frac{1}{\lambda_N^2}),$$

where the second inequality is due to Lemma B.6, Lemma B.7. Therefore, we obtain

$$\mathbb{E}[\|\mathbb{E}[\nabla_x L(x_k, z_k^{2^N-1}(\lambda_N), y_k^{2^N-1}, \lambda_N; \eta_k, \xi_k) \mid \mathcal{F}_k] - u_k(0, \lambda_0)\|^2 \mid \mathcal{F}_k] \leq \mathcal{O}(1).$$

For the second term, we have

$$\mathbb{E}[\|p_n^{-1}(u_k(n, \lambda_n) - u_k(n-1, \lambda_{n-1}))\|^2 \mid \mathcal{F}_k]$$

$$= \sum_{n=1}^{N} p_n^{-1} \mathbb{E}[\|u_k(n, \lambda_n) - u_k(n-1, \lambda_{n-1})\|^2 \mid \mathcal{F}_k] \tag{20}$$

$$\leq \sum_{n=1}^{N} p_n^{-1} \mathcal{O}(\frac{1}{(\lambda_{n-1})^2}) \leq \mathcal{O}(2^{\frac{N}{2}}),$$

where the first inequality is due to Lemma B.14. This completes the proof. $\qquad\square$

The following Lemma estimate the term $\mathbb{E}[\|u_k(n, \lambda_n) - u_k(n-1, \lambda_{n-1})\|^2 \mid \mathcal{F}_k]$ in (20).

**Lemma B.14.** *Under Assumptions 2.1 and 2.3, consider Algorithm 2, we have*

$$\mathbb{E}[\|u_k(n, \lambda_n) - u_k(n-1, \lambda_{n-1})\|^2 \mid \mathcal{F}_k] \leq \mathcal{O}(\frac{1}{2^{\frac{n-1}{2}}}).$$

*Proof.* We denote

$$u_k^{\star}(x_k, \lambda_n; \eta_k, \xi_k) := \nabla_x L(x_k, y^{\star}(x_k, \frac{1}{\lambda_n}; \xi_k), y^{\star}(x_k, 0; \xi_k), \lambda_n; \eta_k, \xi_k).$$

It is easy to verify that

$$u_k(n, \lambda_n) - u_k(n-1, \lambda_{n-1})$$
$$= u_k(n, \lambda_n) - u_k^{\star}(x_k, \lambda_n; \eta_k, \xi_k) + u_k^{\star}(x_k, \lambda_n; \eta_k, \xi_k) - u_k^{\star}(x_k, \lambda_{n-1}; \eta_k, \xi_k) \tag{21}$$
$$+ u_k^{\star}(x_k, \lambda_{n-1}; \eta_k, \xi_k) - u_k(n-1, \lambda_{n-1}).$$

We then analyze the following three terms:

1. $\mathbb{E}[\|u_k(n, \lambda_n) - u_k^{\star}(x_k, \lambda_n; \eta_k, \xi_k)\|^2 \mid \mathcal{F}_k]$;

2. $\mathbb{E}[\|u_k(n-1, \lambda_{n-1}) - u_k^{\star}(x_k, \lambda_{n-1}; \eta_k, \xi_k)\|^2 \mid \mathcal{F}_k]$;

3. $\mathbb{E}[\|u_k^{\star}(x_k, \lambda_n; \eta_k, \xi_k) - u_k^{\star}(x_k, \lambda_{n-1}; \eta_k, \xi_k)\|^2 \mid \mathcal{F}_k]$.

For the first term, we have

$$u_k(n, \lambda_n) - u_k^{\star}(x_k, \lambda_n; \eta_k, \xi_k)$$

$$= \nabla_1 f(x_k, z_k^{2^n-1}(\lambda_n); \eta_k, \xi_k) - \nabla_1 f(x_k, y^{\star}(x_k, \frac{1}{\lambda_n}; \xi_k); \eta_k, \xi_k)$$

$$+ \lambda_n(\nabla_1 g(x_k, z_k^{2^n-1}(\lambda_n); \eta_k, \xi_k) - \nabla_1 g(x_k, y^{\star}(x_k, \frac{1}{\lambda_n}; \xi_k); \eta_k, \xi_k))$$

$$+ \lambda_n(g(x_k, y^{\star}(x_k, 0; \xi_k); \eta_k, \xi_k) - \nabla_1 g(x_k, y_k^{2^n-1}; \eta_k, \xi_k)).$$

Combining the above equality with Lemma B.6, we obtain

$$\mathbb{E}[\|u_k(n, \lambda_n) - u_k^{\star}(x_k, \lambda_n; \eta_k, \xi_k)\|^2 \mid \mathcal{F}_k] \leq \mathcal{O}(\frac{\lambda_n^2}{2^n}). \tag{22}$$

Similarly, we know that

$$\mathbb{E}[\|u_k(n-1, \lambda_{n-1}) - u_k^{\star}(x_k, \lambda_{n-1}; \eta_k, \xi_k)\|^2] \leq \mathcal{O}(\frac{\lambda_{n-1}^2}{2^{n-1}}). \tag{23}$$

Below, we estimate the third term. By Lemma B.2, we can see that

$$\nabla_1 g(x_k, y^{\star}(x_k, \frac{1}{\lambda_n}; \xi_k); \eta_k, \xi_k) - \nabla_1 g(x_k, y^{\star}(x_k, 0; \xi_k); \eta_k, \xi_k)$$

$$= \nabla_{12}^2 g(x_k, y^{\star}(x_k, 0; \xi_k); \eta_k, \xi_k)(y^{\star}(x_k, \frac{1}{\lambda_n}; \xi_k) - y^{\star}(x_k, 0; \xi_k)) + r_1, \tag{24}$$

where $\|r_1\| = \mathcal{O}(\|y^\star(x_k, \frac{1}{\lambda_n}; \xi_k) - y^\star(x_k, 0; \xi_k)\|^2)$, and

$$\nabla_2 g(x_k, y^\star(x_k, 0; \xi_k); \xi_k) - \nabla_2 g(x_k, y^\star(x_k, \frac{1}{\lambda_n}; \xi_k))$$
$$= \nabla_{22}^2 g(x_k, y^\star(x_k, 0; \xi_k); \xi_k)(y^\star(x_k, 0; \xi_k) - y^\star(x_k, \frac{1}{\lambda_n}; \xi_k)) + r_2, \tag{25}$$

where $\|r_2\| = \mathcal{O}(\|y^\star(x_k, \frac{1}{\lambda_n}; \xi_k) - y^\star(x_k, 0; \xi_k)\|^2)$. It follows from Lemma B.7 that $\nabla_\delta y^\star(x, \delta; \xi)$ is Lipschitz continuous, by Lemma B.3, one has

$$y^\star(x, \delta; \xi) - y^\star(x, 0; \xi) = D_\delta y^\star(x, 0; \xi)(\delta - 0) + r_y, \tag{26}$$

where $\|r_y\| = \mathcal{O}(|\delta|^2)$. Combining (26), Lemma B.7 with (24), we can see that

$$u_k^\star(x_k, \lambda_n; \eta_k, \xi_k)$$
$$= \nabla_1 f(x_k, y^\star(x_k, \frac{1}{\lambda_n}; \xi_k); \eta_k, \xi_k) + (\nabla_{12}^2 g(x_k, y^\star(x_k, 0; \xi_k), \eta_k, \xi_k)\nabla_\delta y^\star(x_k, 0, \xi_k) + r_3,$$

where $\|r_3\| = \mathcal{O}(\frac{1}{\lambda_n})$. Similarly, we have

$$u_k^\star(x_k, \lambda_{n-1}; \eta_k, \xi_k)$$
$$= \nabla_1 f(x_k, y^\star(x_k, \frac{1}{\lambda_{n-1}}; \xi_k); \eta_k, \xi_k) + \nabla_{12}^2 g(x_k, y^\star(x_k, 0; \xi_k); \eta_k, \xi_k)\nabla_\delta y^\star(x_k, , 0, \xi_k) + r_4,$$

where $\|r_4\| = \mathcal{O}(\frac{1}{\lambda_{n-1}})$. Therefore, combining the above two equalities with Lemma B.7, It is easy to verify that

$$\|u_k^\star(x_k, \lambda_n; \eta_k, \xi_k) - u_k^\star(x_k, \lambda_{n-1}; \eta_k, \xi_k)\|$$
$$\leq \ell_{f,1}\|y^\star(x_k, \frac{1}{\lambda_n}; \xi_k) - y^\star(x_k, \frac{1}{\lambda_{n-1}}; \xi_k)\| \leq \mathcal{O}(\frac{1}{\lambda_{n-1}}).$$

By the above inequality and (22), (23), one has

$$\mathbb{E}[\|u_k(n, \lambda_n) - u_k(n - 1, \lambda_{n-1})\|^2 \mid \mathcal{F}_k] \leq \mathcal{O}(\frac{\lambda_n^2}{2^n} + \frac{\lambda_{n-1}^2}{2^{n-1}} + \frac{1}{\lambda_{n-1}^2}) \leq \mathcal{O}(\frac{1}{2^{\frac{n-1}{2}}}).$$

This completes the proof. $\qquad\square$

## B.7 PROOF OF THEOREM 3.1

It follows from Lemma B.4 that $F(x)$ is $\ell_{F,1}$-Lipschitz smooth, which implies

$$F(x_{k+1}) - F(x_k) \leq \langle \nabla F(x_k), x_{k+1} - x_k \rangle + \frac{\ell_{F,1}}{2}\|x_{k+1} - x_k\|^2.$$

The above inequality implies

$$\mathbb{E}[F(x_{k+1}) - F(x_k) \mid \mathcal{F}_k]$$
$$\leq \mathbb{E}[\langle \nabla F(x_k), x_{k+1} - x_k \rangle + \frac{\ell_{F,1}}{2}\|x_{k+1} - x_k\|^2 \mid \mathcal{F}_k]$$
$$= -\frac{\alpha_k}{2}(\|\nabla F(x_k)\|^2 + \|\mathbb{E}[\nabla_x L(x_k, z_{k+1}, y_{k+1}, \lambda_k; \eta_k, \xi_k) \mid \mathcal{F}_k]\|^2)$$
$$+ \frac{\alpha_k}{2}\|\nabla F(x_k) - \mathbb{E}[\nabla_x L(x_k, z_{k+1}, y_{k+1}, \lambda_k; \eta_k, \xi_k))] \mid \mathcal{F}_k\|^2 + \frac{\ell_{F,1}}{2}\mathbb{E}[\|x_{k+1} - x_k\|^2 \mid \mathcal{F}_k],$$

where the equality is due to the definition of $x_{k+1}$ in Algorithm 1 and the fact that $\langle a, b \rangle = -\frac{1}{2}(\|a\|^2 + \|b\|^2) + \frac{1}{2}\|a - b\|^2$. For the last term in the above inequality, we have

$$\mathbb{E}[\|x_{k+1} - x_k\|^2 \mid \mathcal{F}_k] \leq 2\alpha_k^2(\ell_{f,0}^2 + \lambda_k^2 \ell_{g,1}^2 \mathbb{E}[\|z_{k+1} - y_{k+1}\|^2 \mid \mathcal{F}_k]),$$

and

$$\mathbb{E}[\|z_{k+1} - y_{k+1}\|^2 \mid \mathcal{F}_k]$$

$$= \mathbb{E}[\|z_{k+1} - y^\star(x_k, \frac{1}{\lambda_k}; \xi) + y^\star(x_k, \frac{1}{\lambda_k}; \xi) - y^\star(x_k, 0; \xi) + y^\star(x_k, 0; \xi) - y_{k+1}\|^2 \mid \mathcal{F}_k]$$

$$\leq 3\mathbb{E}[\|z_{k+1} - y^\star(x_k, \frac{1}{\lambda_k}; \xi)\|^2 \mid \mathcal{F}_k] + 3\mathbb{E}[\|y^\star(x_k, \frac{1}{\lambda_k}; \xi) - y^\star(x_k, 0; \xi)\|^2 \mid \mathcal{F}_k] \qquad (27)$$

$$\quad + 3\mathbb{E}[\|y^\star(x_k, 0; \xi) - y_{k+1}\|^2 \mid \mathcal{F}_k]$$

$$\leq \mathcal{O}(\frac{1}{T_k}) + \mathcal{O}(\frac{1}{\lambda_k^2}) \leq \mathcal{O}(\frac{1}{\lambda_k^2}),$$

where the second inequality follows from Lemma B.6, Lemma B.7. Combining the above three inequalities, we have

$$\mathbb{E}[F(x_{k+1}) - F(x_k) \mid \mathcal{F}_k]$$

$$\leq -\frac{\alpha_k}{2}\mathbb{E}[(\|\nabla F(x_k)\|^2 + \|\mathbb{E}[\nabla_x L(x_k, z_{k+1}, y_{k+1}, \lambda_k; \eta, \xi) \mid \mathcal{F}_k]\|^2)$$

$$\quad + \frac{\alpha_k}{2}\|\nabla F(x_k) - \mathbb{E}[\nabla_x L(x_k, z_{k+1}, y_{k+1}, \lambda_k; \eta, \xi) \mid \mathcal{F}_k]\|^2 + \alpha_k^2 \mathcal{O}(1),$$

which implies

$$\frac{\alpha_k}{2}\mathbb{E}[\|\nabla F(x_k)\|^2 \mid \mathcal{F}_k]$$

$$\leq \frac{\alpha_k}{2}[\|\nabla F(x_k) - \mathbb{E}[\nabla_x L(x_k, z_{k+1}, y_{k+1}, \lambda_k; \eta_k, \xi_k) \mid \mathcal{F}_k]\|^2$$

$$\quad + \mathbb{E}[F(x_k) - F(x_{k+1}) \mid \mathcal{F}_k] + \alpha_k^2 \mathcal{O}(1).$$

Multiply both sides of the above inequality by $\frac{2}{\alpha_k}$, we get

$$\mathbb{E}[\|\nabla F(x_k)\|^2 \mid \mathcal{F}_k]$$

$$\leq \mathbb{E}[\frac{2}{\alpha_k}F(x_k) - \frac{2}{\alpha_{k+1}}F(x_{k+1}) + (\frac{2}{\alpha_{k+1}} - \frac{2}{\alpha_k})F(x_{k+1}) \mid \mathcal{F}_k] + \alpha_k \mathcal{O}(1)$$

$$\quad + \|\nabla F(x_k) - \mathbb{E}[\nabla_x L(x_k, z_{k+1}, y_{k+1}, \lambda_k; \eta_k, \xi_k) \mid \mathcal{F}_k]\|^2.$$

It follows from Lemma B.11 that

$$\|\nabla F(x_k) - \mathbb{E}[\nabla_x L(x_k, z_{k+1}, y_{k+1}, \lambda_k; \eta_k, \xi_k) \mid \mathcal{F}_k]\|^2 \leq \mathcal{O}(\frac{1}{\lambda_k^2}) + \mathcal{O}(\frac{\lambda_k^2}{T_k}).$$

The above two inequalities imply

$$\mathbb{E}[\|\nabla F(x_k)\|^2 \mid \mathcal{F}_k]$$

$$\leq \mathbb{E}[\frac{2}{\alpha_k}F(x_k) - \frac{2}{\alpha_{k+1}}F(x_{k+1}) \mid \mathcal{F}_k] + \mathcal{O}(\frac{1}{\alpha_{k+1}} - \frac{1}{\alpha_k}) + \mathcal{O}(\alpha_k) + \mathcal{O}(\frac{1}{\lambda_k^2}).$$

Therefore, we obtain

$$\frac{1}{K}\sum_{k=1}^{K}\mathbb{E}[\|\nabla F(x_k)\|^2] \leq \mathcal{O}(\frac{1}{\sqrt{K}}).$$

To ensure $\frac{1}{K}\sum_{k=1}^{K}\mathbb{E}[\|\nabla F(x_k)\|^2] \leq \epsilon^2$, it suffices to set $K = \mathcal{O}(\epsilon^{-4})$, $T_K = \mathcal{O}(\epsilon^{-4})$. As a result, the sample complexity of $\nabla_1 f$, $\nabla_1 g$ is of order $\mathcal{O}(\epsilon^{-4})$. The complexity of $\nabla_2 g$, $\nabla_2 f$ is of order $\mathcal{O}(\epsilon^{-8})$.

### B.8   PROOF OF THEOREM 3.4

It follows from Lemma B.4 that $F(x)$ is $\ell_{F,1}$-smooth, which implies

$$F(x_{k+1}) - F(x_k) \leq \langle \nabla F(x_k), x_{k+1} - x_k \rangle + \frac{\ell_{F,1}}{2}\|x_{k+1} - x_k\|^2.$$

For notational simplicity, we adopt the following conventions:

$$v_k(n_k, \lambda_{n_k}) = u_k(0, \lambda_0) + p_{n_k}^{-1}(u_k(n_k, \lambda_{n_k}) - u_k(n_k - 1, \lambda_{n_k-1})). \tag{28}$$

One has

$$\mathbb{E}[F(x_{k+1}) - F(x_k) \mid \mathcal{F}_k]$$

$$\leq \mathbb{E}[\langle \nabla F(x_k), x_{k+1} - x_k \rangle + \frac{\ell_{F,1}}{2}\|x_{k+1} - x_k\|^2 \mid \mathcal{F}_k]$$

$$= \mathbb{E}_{n_k > c_0 N}[-a_1\alpha_0\langle \nabla F(x_k), v_k(n_k, \lambda_{n_k})\rangle + \frac{\ell_{F,1}}{2}\|x_{k+1} - x_k\|^2 \mid \mathcal{F}_k]$$

$$+ \mathbb{E}_{n_k \leq c_0 N}[-\alpha_0\langle \nabla F(x_k), v_k(n_k, \lambda_{n_k})\rangle + \frac{\ell_{F,1}}{2}\|x_{k+1} - x_k\|^2 \mid \mathcal{F}_k],$$

where the equality uses the fact that the expectation of a piece-wise affine function is the sum of expectation of each piece. Subsequent to this, we apply an algebraic manipulation to the right-hand side of the aforementioned inequality to express it in an equivalent form. It follow from (6), (7) that

$$- a_1\alpha_0\langle \nabla F(x_k), \mathbb{E}_{n_k > c_0 N}[v_k(n_k, \lambda_{n_k}) \mid \mathcal{F}_k]\rangle + \frac{\ell_{F,1}}{2}\mathbb{E}_{n_k > c_0 N}[\|x_{k+1} - x_k\|^2 \mid \mathcal{F}_k]$$

$$= - a_1\alpha_0\langle \nabla F(x_k), \mathbb{E}[u_k(N, \lambda_N) - u_k(c_0 N, \lambda_{c_0 N}) \mid \mathcal{F}_k]\rangle + \frac{\ell_{F,1}}{2}\mathbb{E}_{n_k \leq N}[\|x_{k+1} - x_k\|^2 \mid \mathcal{F}_k]$$

$$- \frac{\ell_{F,1}}{2}\mathbb{E}_{n_k \leq c_0 N}[\|x_{k+1} - x_k\|^2 \mid \mathcal{F}_k],$$

and

$$- \alpha_0\langle \nabla F(x_k), \mathbb{E}_{n_k \leq c_0 N}[v_k(n_k, \lambda_{n_k}) \mid \mathcal{F}_k]\rangle + \frac{\ell_{F,1}}{2}\mathbb{E}_{n_k \leq c_0 N}[\|x_{k+1} - x_k\|^2 \mid \mathcal{F}_k]$$

$$= - \alpha_0\langle \nabla F(x_k), \mathbb{E}[u_k(c_0 N, \lambda_{c_0 N}) \mid \mathcal{F}_k]\rangle + \frac{\ell_{F,1}}{2}\mathbb{E}_{n_k \leq c_0 N}[\|x_{k+1} - x_k\|^2 \mid \mathcal{F}_k].$$

Combining the above three equations, we get

$$\mathbb{E}[F(x_{k+1}) - F(x_k) \mid \mathcal{F}_k]$$

$$\leq - a_1\alpha_0\langle \nabla F(x_k), \mathbb{E}[u_k(N, \lambda_N) \mid \mathcal{F}_k]\rangle + \frac{\ell_{F,1}}{2}\mathbb{E}_{n_k \leq N}[\|x_{k+1} - x_k\|^2 \mid \mathcal{F}_k]$$

$$- \alpha_0(1 - a_1)\langle \nabla F(x_k), \mathbb{E}[u_k(c_0 N, \lambda_{c_0 N}) \mid \mathcal{F}_k]\rangle$$

$$= \frac{a_1\alpha_0}{2}\|\nabla F(x_k) - \mathbb{E}[u_k(N, \lambda_N) \mid \mathcal{F}_k]\|^2 - \frac{a_1\alpha_0}{2}\|\nabla F(x_k)\|^2 - \frac{a_1\alpha_0}{2}\|\mathbb{E}[u_k(N, \lambda_N) \mid \mathcal{F}_k]\|^2$$

$$+ \frac{\alpha_0(1 - a_1)}{2}\|\nabla F(x_k) - \mathbb{E}[u_k(c_0 N, \lambda_{c_0 N}) \mid \mathcal{F}_k]\|^2 + \frac{\ell_{F,1}}{2}\mathbb{E}_{n_k \leq N}[\|x_{k+1} - x_k\|^2 \mid \mathcal{F}_k],$$

where the equality is due to the fact that $-\langle a, b \rangle = -\frac{1}{2}(\|a\|^2 + \|b\|^2) + \frac{1}{2}\|a - b\|^2$. For the last term in the above inequality, it is easy to verify that

$$\mathbb{E}_{n_k \leq N}[\|x_{k+1} - x_k\|^2 \mid \mathcal{F}_k]$$

$$= \alpha^2\mathbb{E}_{n_k \leq N}[\|v_k(n_k, \lambda_{n_k})\|^2 \mid \mathcal{F}_k]$$

$$\leq 2\alpha_0^2\mathbb{E}_{n_k \leq N}[\|\nabla F(x_k)\|^2 + \|v_k(n_k, \lambda_{n_k}) - \nabla F(x_k)\|^2 \mid \mathcal{F}_k]$$

$$\leq 2\alpha_0^2\mathbb{E}_{n_k \leq N}[\|\nabla F(x_k)\|^2 + 2\|\mathbb{E}[\nabla_x L(x_k, z_k^{2^N-1}(\lambda_N), y_k^{2^N-1}; \eta_k, \xi_k) \mid \mathcal{F}_k] - \nabla F(x_k)\|^2 \mid \mathcal{F}_k]$$

$$+ 4\alpha_0^2\mathbb{E}_{n_k \leq N}[\|\mathbb{E}[\nabla_x L(x_k, z_k^{2^N-1}(\lambda_N), y_k^{2^N-1}; \eta_k, \xi_k) \mid \mathcal{F}_k] - v_k(n_k, \lambda_{n_k})\|^2 \mid \mathcal{F}_k]$$

$$\leq 2\alpha_0^2\mathbb{E}_{n_k \leq N}[\|\nabla F(x_k)\|^2 \mid \mathcal{F}_k] + 4\alpha_0^2\mathcal{O}(\frac{1}{\lambda_N^2}) + 4\alpha_0^2\mathcal{O}(2^{\frac{N}{2}}),$$

where the first equality is due to the definition of $x_{k+1}$, the first and second inequalities follow from the triangle inequality, and the last inequality follows from Lemma B.12, Lemma B.13.

Combining the above two inequalities, and then taking the expectation on the new inequality, if $\alpha_0 \leq \frac{a_1}{8\ell_{F,1}}$, one has

$$\mathbb{E}[F(x_{k+1}) - F(x_k)]$$

$$\leq \frac{a_1\alpha_0}{2}\mathbb{E}[\|\nabla F(x_k) - \mathbb{E}[u_k(N, \lambda_N) \mid \mathcal{F}_k]\|^2] + (2\alpha_0^2\ell_{F,1} - \frac{a_1\alpha_0}{2})\mathbb{E}[\|\nabla F(x_k)\|^2]$$

$$+ \frac{\alpha_0(1-a_1)}{2}\mathbb{E}[\|\nabla F(x_k) - \mathbb{E}[u_k(c_0N, \lambda_{c_0N}) \mid \mathcal{F}_k]\|^2] + \alpha_0^2\mathcal{O}(\frac{1}{\lambda_N^2}) + \alpha_0^2\mathcal{O}(2^{\frac{N}{2}})$$

$$\leq \frac{a_1\alpha_0}{2}\left(\mathcal{O}(\frac{1}{\lambda_N^2}) + \mathcal{O}(\frac{\lambda_N^2}{2^N})\right) - \frac{a_1\alpha_0}{4}\mathbb{E}[\|\nabla F(x_k)\|^2]$$

$$+ \frac{\alpha_0(1-a_1)}{2}\left(\mathcal{O}(\frac{1}{\lambda_{c_0N}^2}) + \mathcal{O}(\frac{\lambda_{c_0N}^2}{2^{c_0N}})\right) + \alpha_0^2\mathcal{O}(\frac{1}{\lambda_N^2}) + \alpha_0^2\mathcal{O}(2^{\frac{N}{2}}),$$

where the last inequality is due to Lemma B.8, Lemma B.10 and $\alpha_0 \leq \frac{a_1}{8\ell_{F,1}}$ (which implies $2\ell_{F,1}\alpha_0^2 - \frac{a_1\alpha_0}{4} \leq 0$). Therefore, we get

$$\mathbb{E}[\|\nabla F(x_k)\|^2] \leq \frac{4}{a_1\alpha_0}(\mathbb{E}[F(x_k)] - \mathbb{E}[F(x_{k+1})]) + \mathcal{O}(\frac{1}{\lambda_N^2} + \frac{\lambda_N^2}{2^N})$$

$$+ \mathcal{O}(\frac{1}{\lambda_{c_0N}^2} + \frac{\lambda_{c_0N}^2}{2^{c_0N}}) + \alpha_0\mathcal{O}(\frac{1}{\lambda_N^2}) + \alpha_0\mathcal{O}(2^{\frac{N}{2}}).$$

The above inequality and the definition of $\lambda_{n_k}$ in Algorithm 2 imply

$$\frac{1}{K}\sum_{k=1}^{K}\mathbb{E}[\|\nabla F(x_k)\|^2] \leq \frac{4\mathbb{E}[F(x_1) - F(x_{K+1})]}{a_1\alpha_0 K} + \mathcal{O}(\frac{1}{\lambda_N^2} + \frac{1}{\lambda_{c_0N}^2}) + \alpha_0\mathcal{O}(\frac{1}{\lambda_N^2} + 2^{\frac{N}{2}})$$

The average number of iterations required for the inner loop is

$$\sum_{n_k=1}^{N}(2^{n_k+1} - 1)\frac{2^{-n_k}}{1 - 2^{-N-1}} < 3N.$$

To ensure $\frac{1}{K}\sum_{k=1}^{K}\mathbb{E}[\|\nabla F(x_k)\|^2] \leq \epsilon^2$, it suffices to set $\alpha_0 = \mathcal{O}(1)\epsilon^4$, $K = \mathcal{O}(\epsilon^{-6})$, $N = \mathcal{O}(1)\log(\epsilon^{-1})$. As a result, the sample complexity of $\nabla_1 f, \nabla_1 g$ is of order $\mathcal{O}(\epsilon^{-6})$. The complexity of $\nabla_2 g, \nabla_2 f$ is of order $\mathcal{O}(\epsilon^{-6}\log(\epsilon^{-1}))$.

### B.9  THE SETTING OF NUMERICAL EXPERIMENT (META-LEARNING)

We tune the algorithm parameters of these four methods to make sure every method works well: we set $\ell_{f,1} = \mu_g = 1000$; for Algorithm 1, we use $\alpha_k = 25/\sqrt{k+1}$, $\beta_t = 500/(\mu_g(t+1))$ and $K = 1500$; for Algorithm 2, we use $\epsilon = 1e - 4$ and so $N = 4\log(\epsilon^{-1}) \approx 37$, $\alpha_0 = 1$, $c_0N = 10$, $a_1 = 0.05$, $\beta_t = 25/(\mu_g(t+1))$ and $K = 17000$; for Hessian-based method, we follow the settings in Hu et al. (2023b) and use maximum iterations 10000, the RT-MLMC level $K = 12$, $L_{g,1} = 10$, $\alpha_t = 0.5/\sqrt{t}$ for $t <= 1000$ and $0.5/t$ for $t > 1000$, the stepsize for the inner update is replaced by $\beta_t = 70/(t+1)$ rather than $70/2^t$ for better performance; for the reduction method in Bouscary et al. (2025), we use basis degrees 50, for stocBiO for solving the reduced SBO problem, we use maximum iterations 3000, inner iterations $D = 100$, stepsizes $\alpha = 0.01$, $\beta = 0.1$ and $\eta = 1e - 3$, and length of Neumann series $Q = 30$. To handle the high variance of Algorithm 2 and RT-MLMC Hessian-based methods, we use minibatch over the hypergradient estimators for the outer loop. Specifically, for Algorithm 2, in the $k$-th outer iteration, given $x_k$, we sample an $n_k$ from the truncated geometric distribution, and then repeat steps 3 to 13 in Algorithm 2 for 10 times.

Similarly, for RT-MLMC Hessian-based method, in the $k$-th outer iteration, we sample a $\hat{k}$ from the truncated geometric distribution, and then repeat `EpochSGD` (c.f., Algorithm 1 in Hu et al. (2023b) for 10 times to compute the averaged gradient estimator to update $x_{k+1}$. Note that the maximum number of iterations are set to ensure that the computational time for these three algorithms is roughly comparable.

## B.10 THE SETTING OF NUMERCAL EXPERIMENT (WDRO-SI)

In this experiment, $\xi, y \in \mathbb{R}^{100}$, $\gamma_1 = 10$, and the parameters in $l_\beta$ are set to $h = 1$, $b = 5$ and $\beta = 5$. We use a three-layer fully-connected neural network as the mapping $f(x; \cdot)$, where the neurons in each layer are $[64, 32, 1]$, the activation functions of hidden layers are ReLU, and the output layer uses the sigmoid function scaled by 10. To construct the nominal distribution, we first uniformly randomly generate the true $x^*$, and $M = 50$ contexts $\{\xi_i\}_{i=1}^M$. For each $\xi_i$, we generate $\{\eta_j = f(x^*; \xi_i) + \epsilon\}_{j=1}^{100}$ with $\epsilon$ being white noise. The performance is evaluated by the stationarities and expected losses $\mathbb{E}_{(\xi, \eta) \sim P^0}[l(f(x; \xi), \eta)]$, where the expectation is approximated using sample average over $20,000$ sample points $\{(\xi_i, \eta_i)\}_{i=1}^{20,000}$ that are generated using the same scheme as the training nominal distribution. Similarly to the meta-learning example, these losses are evaluated only on the 50 equally spaced grid points.

The algorithm parameters of each method are tuned to ensure the good performance. Specifically, we set $\ell_{f,1} = \mu_g = 1000$; for Algorithm 1, we use $\alpha_k = 0.5/\sqrt{k+1}$, $\beta_t = 5/(\mu_g(t+1))$ and $K = 100$; for Algorithm 2, we use $\epsilon = 1e-4$ and so $N = 4\log(\epsilon^{-1}) \approx 37$, $\alpha_0 = 0.5$, $c_0 N = 10$, $a_1 = 0.05$, $\beta_t = 1/(\mu_g(t+1))$ and $K = 1,000$; for Hessian-based method, we use maximum iterations $1,000$, the RT-MLMC level $K = 12$, $L_{g,1} = 10$, $\alpha_t = 1e-5/\sqrt{t}$ for $t <= 1000$ and $1e-5/t$ for $t > 1000$, the stepsize for the inner update is replaced by $\beta_t = 5e-5/(t+1)$; for the reduction method in Bouscary et al. (2025), we use basis degrees 5, for stocBiO for solving the reduced SBO problem, we use maximum iterations 300, inner iterations $D = 100$, stepsizes $\alpha = 0.01$, $\beta = 0.01$ and $\eta = 1e-4$, and the length of Neumann series $Q = 30$. Different from the meta-learning example, we do not use minibatch for RT-MLMC methods.