# OpenReview forum: "Fully First-order Methods for Contextual Stochastic Bilevel Optimization"
_ICLR.cc/2026/Conference — Submitted to ICLR 2026_

### Official Review · Reviewer_kauD · 2025-10-31

**Soundness:** 2
**Presentation:** 3
**Contribution:** 2
**Rating:** 4
**Confidence:** 3

**Summary:**

The paper studies contextual stochastic bilevel optimization (CSBO) and proposes a double-loop, fully first-order algorithm with sample/gradient complexities ($\tilde{\mathcal O}(\epsilon^{-8})$). An accelerated variant, built on random truncated multilevel Monte Carlo, improves this to $(\tilde{\mathcal O}(\epsilon^{-6}))$.

**Strengths:**

1.This paper proposes a fully first-order algorithm that does not rely on second-order information.

2.The numerical experiments demonstrate the empirical performance of the proposed methods.

3.This paper includes a helpful roadmap that improves readability.

**Weaknesses:**

1.Despite the authors’ emphasis that the contribution is fully first-order, the reported complexity results are not fully satisfactory. To my knowledge, for classical stochastic bilevel optimization, Chen et al. [1] proposed a near-optimal fully first-order method whose rate is within only a logarithmic factor of second-order methods; meanwhile, Bouscary et al. [2] established a connection between CSBO and SBO and—though using second-order information—achieved a sampling complexity of $(\tilde{\mathcal O}(\epsilon^{-3}))$. These comparisons may indicate that there could still be some room to further tighten the results.

2. The method adopts a double-loop architecture, which might introduce some implementation overhead in practice and could limit scalability and applicabilit.

3. The analysis requires a strongly convex lower-level problem.

[1]Chen L, Ma Y, Zhang J. Near-optimal nonconvex-strongly-convex bilevel optimization with fully first-order oracles[J]. Journal of Machine Learning Research, 2025, 26(109): 1-56.

[2]Bouscary M, Zhang J, Amin S. Reducing Contextual Stochastic Bilevel Optimization via Structured Function Approximation[J]. arXiv preprint arXiv:2503.19991, 2025.

**Questions:**

1.In Algorithm 1, line 2, the stepsize choice for $(\alpha)$ includes $(\mathcal{O}(1))$. Is there a specific reason to emphasize $(\mathcal{O}(1)) $ here, or would a fixed constant suffice?

2.Related to Weakness 1: given the current results, it might be helpful to comment on whether sharper bounds could be attainable, or to offer some intuition on potential barriers to improving the rates under the present setting.

Minor typo: Line 444, “eary” --> “early”.

---

> ### Author Response · Authors · 2025-11-21
> **Thank you for your comments.**
>
> ## Q1: Complexity
> We would like to highlight that [Chen et. al 2024] and its journal version [Chen et. al 2025 (JMLR)] achieves $O(\epsilon^{-4})$ complexity when only one level has randomness, which is a partial stochastic setting that only considered in its paper. When both levels have randomness as considered in our work and other bilevel works, they achieve $O(\epsilon^{-6})$ complexity for the noncontextual setting. Our accelerated method also achieves the same $O(\epsilon^{-6})$ complexity for the clearly more complicated contextual settings.
>
> Second, we would like to clarify that even in the non-contextual regime, fully first-order methods are slower than second-order methods in terms of complexity. The current known fastest first-order method for fully stochastic bilevel optimization (SBO) problem (e.g., stochasicity appears on both upper level and lower level problems) is the one proposed in [Chen et. al 2024], which achieves complexity $O(\epsilon^{-6})$ that is larger than $O(\epsilon^{-4})$ achieved by second-order method. When incorporating variance reduction, also note that [Kwon et. al 2023] proposes a first-order method for SBO which achieves complexity $O(\epsilon^{-5})$. In contrast, the fastest second-order methods for SBO achieve complexity $O(\epsilon^{-3})$, which matches the lower complexity of SBO [Khandur et. al 2021].
>
> While the sample complexity is higher, fully first-order methods have a much lower per-iteration cost compared to second-order algorithms, making them suitable for large-scale problems. For example, consider the meta-learning problem in section 4, computing the gradient costs O(d) flops, and computing the Hessian costs O(d^2) flops. In total, the per-iteration flops cost of our Algorithm 1 and Algorithm is $O(T_kd_y+d_x)$, while it is $O(Nd_y^2+d_xd_y+T_kd_y)$ in [Hu et. al 2023] with second-order methods. Thus we see that our methods are much faster in wall-clock time.
> ## Q2: Double loop
> We agree that double loop could be more complicated. However, we would also like to highlight that the context $\xi$ makes it very difficult to build single-loop methods as in the noncontextual setting. In fact, double loop method could sometime be easier to tune as the inner loop already converges as in Algorithm 1. This would lead to low bias low variance hypergradient estimators, making tuning for outer loop easy. For Algorithm 2 the inner loop only runs $\log(\epsilon^{-1})$ steps on average and $O(1)$ steps in most cases, which is cheap and similar to single-loop methods.
> ## Q3: Strong convexity
> Strong convexity is a common assumption in the literature. Extending the analysis to PL-type conditions on the lower-level problem is indeed an interesting direction. If the lower level still has unique optimal solution, our theory still applies. However, we would like to point out that the lower-level problem under the PL condition could admit multiple optimal solutions. This leads to a modelling question, which optimal solution is used in defining the upper level objective, i.e., usually one could take the optimistic (pessimistic) one that ensures the smallest (largest) upper level objective. Then the problem naturally becomes a three-level problem. In addition, the optimality set of the lower-level problem can be nonconvex. Then the introduced middle level is optimizing over a nonconvex set, which itself can be super challenging as well.  We will add the corresponding discussion and leave it for future work, i.e., interested readers might refer to  “Set Smoothness Unlocks Clarke Hyper-stationarity in Bilevel Optimization” for the noncontextual setting.
> ## Q4
> Thanks for your question. We just want to highlight that it is a constant independent of accuracy $\epsilon$. We will make it to be more concise in the revision.
> ## Q5
> It is of course interesting to obtain lower bounds. However, we would like to highlight that our accelerated method already matches the best fully first-order algorithm for noncontextual bilevel setting [Chen et al., 2025 JMLR]. Thus to study lower bounds, the contextual and noncontextual problems should be studied together to see if fully-first order regime is fundamentally harder than classical nonconvex optimization. In fact, building an oracle model to analyse fully first-order methods could be very challenging, which remains open even in the noncontextual setting. On the other hand, one might also explore additional assumption or variance reduction techniques like SPIDER to further accelerate. All of these are for future studies.
> ## Minor typo
> Thank you for pointing it out.

---

### Official Review · Reviewer_DoJP · 2025-10-31

**Soundness:** 3
**Presentation:** 2
**Contribution:** 2
**Rating:** 4
**Confidence:** 4

**Summary:**

This paper studies contextual stochastic bilevel optimization (CSBO), where the lower-level problem depends on both the upper-level variable $x$ and a context $\xi$. The authors introduce the first fully first-order (Hessian-free) double-loop method for CSBO. The approach uses a penalty formulation $L(x, z, y, \lambda)$ (Eq. 3) and solves two inner stochastic problems $Q(x, y, \delta; \xi)$ (Eq. 4) at $\delta = 0$ and $\delta = 1/\lambda$. The resulting estimator avoids Hessian-inverse computations entirely.

Two algorithms are proposed: Algorithm 1: Basic first-order CSBO with $\tilde{O}(\varepsilon^{-8})$ complexity (Theorem 3.1). Algorithm 2: Accelerated variant leveraging Random Truncated Multi-Level Monte Carlo (RT-MLMC) and an adaptive step-size gate, achieving $\tilde{O}(\varepsilon^{-6})$ complexity (Theorem 3.4).

Experiments on meta-learning tasks constructed from TinyImageNet embeddings show that Algorithm 2 attains faster convergence than Hessian-based baselines and demonstrates robustness to variance growth caused by large $\lambda$ (Figs. 1–4).

**Strengths:**

1. Innovative first-order surrogate: A well-motivated penalty objective and twin inner SGD formulation (Eqs. 3–5, Alg. 1).

2. Variance-aware acceleration: RT-MLMC with adaptive gate stabilizes noisy gradients effectively (Alg. 2, Lemma 3.2, Theorem 3.4).

3. Transparent theory: Clear assumptions, proof structure, and complexity results.

4. Empirical validation: Algorithm 2 achieves faster wall-clock convergence than Hessian-based CSBO (Figs. 2–3).

**Weaknesses:**

1. Limited experimental breadth: Evaluations focus solely on meta-learning tasks using linear models and synthetic contexts.

2. Ablations missing: No analysis on $\lambda$ growth rate, truncation level $N$, or adaptive gate parameters $(c_0, a_1)$.

3. Cost per iteration under-reported: Lacks comparison of wall-clock time per gradient or memory usage between Algorithms 1–2 and Hessian-based methods.

4. Strong convexity requirement: May not hold for many practical CSBO tasks; extending to PL-condition or weak convexity would increase applicability.

5. Open gap to optimal rate: The variance bottleneck ($\tilde{O}(\varepsilon^{-2})$) prevents achieving $\tilde{O}(\varepsilon^{-4})$ complexity.

6. Incomplete statistical reporting: Missing standard deviations in all figures and convergence tables.

**Questions:**

1. How sensitive is performance to the $\lambda_k$ schedule? Could it be adaptively adjusted using inner residuals?

2. How were gate parameters $(c_0, a_1)$ selected in Algorithm 2? Is there a theoretical guideline or heuristic tuning rule?

3. Can the RT-MLMC estimator be further variance-reduced (e.g., via control variates) to reach $\tilde{O}(\varepsilon^{-4})$ complexity?

4. Would your theory still hold if $g(x, y; \xi)$ satisfies only the Polyak-Łojasiewicz (PL) condition instead of strong convexity?

5. How does Algorithm 1 perform when contexts $\xi$ are discrete and repetitive (e.g., finite meta-tasks)?

---

> ### Author Response · Authors · 2025-11-21
> **Thank you for your comments**
>
> ## Q1: Experiments
> Thank you for your comments.  In the revised manuscript, we have incorporated the following to strengthen our numerical section:
> 1. We add an experiment of WDRO-SI.
> 2. We  compare our methods with the reduction method in [1], with the reduced problem solved by stocBiO in [2].
> 3. We increase the problem scale of the meta-learning example by increasing the number of classes and tasks.
> ## Q2: Parameter selection
> For the analysis for $\lambda$ and $N$, see Lemma B.8 and Section B.8 Proof of Theorem 3.4. In particular, $N=O( \log (\epsilon^{-1}))$. The setup of $\lambda$ follows Line 2 in Algorithm 1 and Line 4 in Algorithm 2. As for $c_0 \in (0,1)$, $a_1\in (0,1)$, these constants do not affect the analysis. During implementation, they can be tuned within the range $(0,1)$.
> ## Q3: Wall-clock and Memory
> Thank you for pointing it out. In the revised manuscript, we will compare wall-clock time and memory of **per-hypergradient estimation** between our algorithms 1&2 and the Hessian-based methods in [3]. In short, the wall-clock times for algos 1&2 are not directly comparable because the number of inner iterations of algo1 increases along with the outer iteration, which makes the per-hypergradient estimation wall-clock time of algo1 increases along with the outer iteration. In contrast, the averaged time of algo2 is unchanged for all iterations. A simple example is, for metalearning example, when algo1 uses $K=1000$ and algo2 uses $K=14000$, algo2 estimates a hyper-gradient about **13 times** faster than Algorithm 1. Algorithms 1&2 have a much smaller memory cost than the Hessian-based algorithm.
>
> ## Q4: PL condition in the lower level
> Extending the analysis to PL-type conditions on the lower-level problem is indeed an interesting direction. If the lower level still has unique optimal solution, our theory still applies. However, we would like to point out that the lower-level problem under the PL condition could admit multiple optimal solutions. This leads to a modelling question, which optimal solution is used in defining the upper level objective, i.e., usually one could take the optimistic (pessimistic) one that ensures the smallest (largest) upper level objective. Then the problem naturally becomes a three-level problem. In addition, the optimality set of the lower-level problem can be nonconvex. Then the introduced middle level is optimizing over a nonconvex set, which itself can be super challenging as well.  We will add the corresponding discussion and leave it for future work, i.e., interested readers might refer to  “Set Smoothness Unlocks Clarke Hyper-stationarity in Bilevel Optimization” for the noncontextual setting.
> ## Q5; Achieving $O(\epsilon^{-4})$
> You are right that the high variance prevents achieving $O(\epsilon^{-4})$. We believe that this is mainly due to the increasing penalty variable that introduces additional variance as the slater condition does not hold. To address this, one might need to pose additional higher-order smoothness assumptions or incorporate variance reduction like SPIDER or SARAH as you suggested in Q9. We anticipate this would lead to $O(\epsilon^{-4})$ or  $O(\epsilon^{-5})$ complexity, which would be a very interesting open question.
> ## Q6: Standard deviation
> Figures 1-3 show the confidence region (the shaded bands for each curve), which serves the same purpose of illustrating the stability or variance of the algorithms. We are happy to include standard deviation in the revised version.
> ## Q7
> We choose $\lambda_k=  \frac{2\ell_{f,1}}{\mu_g}(k+1)^{1/4}$ in Algorithm 1 line 2, $\lambda_k= \frac{2\ell_{f,1}}{\mu_g}(2^{n_k})^{\frac{1}{4}}$ in Algorithm 2 line 4 as suggested by the theoretical results. These choice directly give good empirical performance without tuning and thus we keep to it. We will test more tuning in the experiments.
> ## Q8: Setup of $c_0$, $a_1$
> Please see Q2. These are constants that one could choose between $(0,1)$.
> ## Q9
> Please see Q5.
> ## Q10
> Please see Q4.
> ## Q11
> Our analysis is quite general; it encompasses the scenario where the $\xi$ is continuous or is discrete and repetitive. In cases where the support size of $\xi$ is particularly small, we suggest to further incorporate warm start (like in the noncontextual setting) for each realization in Algorithm 1 to accelerate lower level computation.
> ## References
> [1] Maxime Bouscary, Jiawei Zhang, and Saurabh Amin. Reducing contextual stochastic bilevel optimization via structured function approximation. arXiv preprint arXiv:2503.19991, 2025.
>
> [2] Kaiyi Ji, Junjie Yang, and Yingbin Liang. Bilevel optimization: Convergence analysis and enhanced design. In Proceedings of the 38th International Conference on Machine Learning, volume 139 of Proceedings of Machine Learning Research, pages 4882–4892. PMLR, 18–24 Jul 2021.
>
> [3] Yifan Hu, Jie Wang, Yao Xie, Andreas Krause, and Daniel Kuhn. Contextual stochastic bilevel optimization. NeurIPS 2023.

---

### Official Review · Reviewer_AtLm · 2025-10-31

**Soundness:** 2
**Presentation:** 3
**Contribution:** 2
**Rating:** 4
**Confidence:** 4

**Summary:**

This paper studied contextual stochastic bilevel optimization and proposed a fully first order approach to solve it. To enhance the efficiency, they proposed an enhanced algorithm based on RT-MLMC technique which reduced the variance of gradient estimator by important sampling. Theoretical analysis is provided for both methods and numerical experiments validate the effectiveness of the proposed methods.

**Strengths:**

This is the first work proposing the first order method for contextual stochastic bilevel optimization with theoretical guarantee.

**Weaknesses:**

1. The major concern is that the theoretical analysis for the proposed methods seems not tight compared with either non-contextual stochastic bilevel optimization (${\cal O}(\epsilon^{-4})$ in [2]) or second-order contextual stochastic bilevel methods (${\cal O}(\epsilon^{-4})$ in [3]). Also, it is unclear whether the gap arises from the analysis or from inherent properties of the contextual setting. It would strengthen the paper to include a lower-bound analysis or to explain why the contextual formulation necessarily yields slower rates (e.g., due to additional variance terms, conditioning, or dependence on context complexity).
2. Another concern is the limited experimental scale. This paper only tested one meta-learning application, which is insufficient to establish broader effectiveness. Moreover, a comparison to methods in the same setting is missing—such as [1], which is applicable when the contextual parameter space is finite.

[1] Maxime Bouscary, Jiawei Zhang, and Saurabh Amin. Reducing contextual stochastic bilevel optimization via structured function approximation. arXiv preprint arXiv:2503.19991, 2025.
[2] Lesi Chen, Jing Xu, and Jingzhao Zhang. On finding small hyper-gradients in bilevel optimization: Hardness results and improved analysis. CoLT, 2024.
[3] Yifan Hu, Jie Wang, Yao Xie, Andreas Krause, and Daniel Kuhn. Contextual stochastic bilevel optimization. NeurIPS 2023.

**Questions:**

1. In theorem, Algorithm 2 has better convergence rate but in experiments (Figure 1), algorithm 1 is faster with respect to the outer iteration.   Could you get some insights from the theoretical analysis that in which setting, Algorithm 2 tends to outperform?

2. I'm wondering the key reason for slower convergence rate. In non-contextual bilevel optimization, the rate of first order bilevel method matches that of second-order bilevel methods.

3. In Remark 3.3, I did not understand why the variance will be ${\cal O}(\epsilon^{-2})$? When $N=c\log(\epsilon^{-1})$, it seems that $\mathcal{O}(2^{\frac{N}{2}})={\mathcal{O}}\left(2^{\log(\epsilon^{-c/2})}\right)=\widetilde{\mathcal{O}}\left(\epsilon^{-c/2}\right)$ which suggests the final complexity depends on the choice of ${\cal O}(1)$ constant c?

---

> ### Author Response · Authors · 2025-11-21
> **Thank you for your comments**
>
> ## Response: Theoretical Analysis
> **Comparison with Non-contextual SBO Algorithms:**   The contextual setting is significantly more complicated than its non-contextual counterpart. Consequently, algorithms designed for non-contextual problems, such as those employing warm-start strategies, fail to generalise. This is because the optimal lower-level solution in the contextual case depends not only on $x$ but also on the context  $\xi$ and thus requires new algorithmic design, as we did in our work. We would like to highlight that [2] and its journal version [3] achieves $O(\epsilon^{-4})$ complexity when only one level has randomness. When both levels have randomness as considered in our work, they achieve $O(\epsilon^{-6})$ complexity for the noncontextual setting. Our accelerated method also achieves $O(\epsilon^{-6})$ complexity for the clearly more complicated contextual setting.
>
> **Comparison with Second-order CSBO Algorithms:** Note that for the nonconvextual setting, the fully first-order method in [2] achieves $O(\epsilon^{-6})$ whereas the second order method could achieve $O(\epsilon^{-4})$. Thus we believe for fully first-order method to get $O(\epsilon^{-4})$, it might require additional assumptions on higher-order smoothness. For instance, a very recent paper, that came out during the month of the ICLR deadline, demonstrates it for the noncontextual setting; see [4]. We will add relevant discussion in the revised version. Despite slower sample complexity compared to second-order methods, we would like to highlight that our fully first-order methods have lower per-iteration cost, making them suitable for large-scale problems. For example, consider the meta-learning problem in section 4 numerical experiments, we can see that the per-iteration flops cost of our Algorithm 1 and Algorithm 2 is $O(T_kd_y+d_x)$, while it is $O(Nd_y^2+d_xd_y+T_kd_y)$ in [5]. Thus for large $d_y$, our methods have clear advantages in terms of wall-clock runtime.
> ## Response: Experiements
> Thank you for your comments. In the revised manuscript, we have incorporated the following to strengthen our numerical section:
> 1. We add an experiment of Wasserstein Distributionally Robust Optimization with side information (WDRO-SI).
> 2. We  compare our methods with the reduction method in [1], with the reduced problem solved by stocBiO in [6].
> 3. We increase the problem scale of the meta-learning example by increasing the number of classes and tasks.
> ## Q1
> The total computational time can be divided into two parts, the number of iterations and the per-iteration computational costs. As demonstrated in the analysis, Algorithm 2 generally requires more iterations than Algorithm 1 as the variance of the gradient estimator obtained from the inner loop of Algorithm 2 is bigger than that of Algorithm 1. Nevertheless, on average, the per-iteration computational cost of Algorithm 2 is significantly smaller than Algorithm 1. Hence in total, Algorithm 2 outperforms Algorithm 1 in terms of the wall-clock time.
> ## Q2
> As discussed earlier, even in the non-contextual bilevel setting, when both levels have randomness, the fully first-order method in [2] achieves $O(\epsilon^{-6})$ whereas the second order method could achieve $O(\epsilon^{-4})$. [2] indeed can achieve $O(\epsilon^{-4})$ but when only one level has randomness, which is not a fair comparison. Back to your question, in both (non)contextual settings, fully first-order methods is slower in terms of convergence compared to second-order methods. We believe that this is mainly due to the increasing penalty that introduces additional variance as the Slater condition does not hold. To address this, one might need to pose additional higher-order smoothness assumptions and etc.
> ## Q3
> Thank you for pointing it out. We will make it clear in the revised version. Note that the selection of $N$ depends on how good we solve the lower-level problem to $\epsilon$ optimality. Thus we pick $N$ such that $2^{N/2} = \epsilon^{-2}$, thus $N=4\log(\epsilon^{-1})$, meaning $c=4$.
> ## References
> [1] Maxime Bouscary, Jiawei Zhang, and Saurabh Amin. Reducing contextual stochastic bilevel optimization via structured function approximation. arXiv:2503.19991, 2025.
>
> [2] Lesi Chen, Jing Xu, and Jingzhao Zhang. On finding small hyper-gradients in bilevel optimization: Hardness results and improved analysis. CoLT, 2024.
>
> [3] Lesi Chen, Yaohua Ma, and Jingzhao Zhang. Near-optimal nonconvex-strongly-convex bilevel optimization with fully first-order oracles. Journal of Machine Learning Research, 26(109):1–56, 2025.
>
> [4] Chen, Lesi, Junru Li, and Jingzhao Zhang. Faster Gradient Methods for Highly-smooth Stochastic Bilevel Optimization. arXiv preprint arXiv:2509.02937 (2025).
>
> [5] Yifan Hu, Jie Wang, Yao Xie, Andreas Krause, and Daniel Kuhn. Contextual stochastic bilevel optimization. NeurIPS 2023.
>
> [6] Kaiyi Ji, Junjie Yang, and Yingbin Liang. Bilevel optimization: Convergence analysis and enhanced design. ICLR 2021.

---

### Official Review · Reviewer_rngr · 2025-11-05

**Soundness:** 3
**Presentation:** 3
**Contribution:** 3
**Rating:** 6
**Confidence:** 3

**Summary:**

The paper addresses contextual stochastic bilevel optimization, where the lower solution depends on both the upper variable and a context drawn from a possibly uncountable set. It proposes fully first-order algorithms based on a penalty reformulation and establishes sample and gradient complexities.

**Strengths:**

The motivation is clear since this is positioned as the first fully first-order treatment of general CSBO that does not rely on second-order oracles. Precise analysis is given under different conditions.

**Weaknesses:**

The proven complexities are weaker than accelerated Hessian based CSBO in terms of epsilon, and the paper notes that direct rate comparison is delicate due to per iteration costs.

**Questions:**

Do you believe that a fully first-order approach in contextual stochastic bilevel optimization can be competitive with second-order methods? I wonder if the proven complexities can be better or not.

On the other hand, if we compare that in wall-clock time, then is it possible to please specify regimes where a fully first-order approach is better by quantifying gradient and Hessian costs, problem dimension, and target accuracy, etc?

---

> ### Author Response · Authors · 2025-11-21
> **Thank you for your comments.**
>
> Yes, we believe that a fully first-order algorithm will eventually be faster than a second-order one. This is based on two observations: First, computing the gradient costs O(d) flops while computing the Hessian costs O(d^2) flops for a function from $R^d$ to $R$. In total, the per-iteration flops cost of our Algorithm 1 and Algorithm 2 is $O(T_kd_y+d_x)$, while it is $O(Nd_y^2+d_xd_y+T_kd_y)$ in [Hu et. al 2023] that uses second-order information. For lower-level problems with higher dimensions, a fully first-order method would have significant accelerations.  Experimental results in Section 4 show a clear advantage in runtime for fully first-order methods. In terms of the proved sample and iteration complexity, this could be further improved by combining variance reduction techniques like SPIDER/SARAH, which we anticipate to get $O(\epsilon^{-4})$ or $O(\epsilon^{-5})$ complexity. We leave this for future exploration.
>
> [Hu et. al 2023] Yifan Hu, Jie Wang, Yao Xie, Andreas Krause, and Daniel Kuhn. Contextual stochastic bilevel optimization. NeurIPS 2023.

---

### Meta-Review · Area_Chair_fknc · 2026-01-05

**Summary:**

This paper proposes the first fully first-order algorithm for Contextual Stochastic Bilevel Optimization (CSBO), a challenging problem with a semi-infinite nature. By incorporating a Random Truncated Multilevel Monte Carlo (RT-MLMC) technique, the authors develop an accelerated version that achieves a sample complexity of $\tilde{\mathcal{O}}(\epsilon^{-6})$, avoiding computationally expensive Hessian evaluations.

Reviewers generally acknowledge the novelty of this work as the first Hessian-free method for general CSBO. However, initial scores were borderline (6, 4, 4, 4), driven by two primary concerns. Primary concerns, raised by Reviewers rngr, AtLm, DoJP and kauD, focused on the gap in theoretical complexity compared to second-order methods. Meanwhile, Reviewers DoJP and AtLm highlighted the limited breadth of experiments, noting that the original submission contained only meta-learning tasks.

In their rebuttal, the authors attempted to clarify that the theoretical gap stems from different problem settings and emphasized the per-iteration computational advantages of their first-order approach. They also expanded the evaluation by including Wasserstein Distributionally Robust Optimization (WDRO-SI) tasks. Nevertheless, the consensus among reviewers remains that the convergence rate of $\tilde{\mathcal{O}}(\epsilon^{-6})$ represents a substantial theoretical regression compared to the $\mathcal{O}(\epsilon^{-4})$ baseline of second-order methods. While the authors argue this is optimal for the fully stochastic setting, the slow convergence rate significantly limits the algorithmic impact. Although WDRO-SI experiments were added, the comparison is flawed. The Hessian-based baseline in Figure 6 exhibits almost zero convergence (a flat line) due to improper tuning (restricted stepsize to avoid instability), while the proposed method shows high variance. Beating a paralyzed baseline does not constitute strong evidence.

Given that three out of four reviewers maintained borderline-reject scores, the paper is not yet ready for publication. Rejection is recommended.

**Reviewer Concerns:**

**Addressed**:

The authors provided references to clarify that the $\mathcal{O}(\epsilon^{-4})$ rate in [1] typically applies to partially stochastic settings, distinguishing their fully stochastic setting.

The authors included the requested comparison with the reduction-based method [2] combined with stocBiO, technically satisfying Reviewer AtLm's request for broader baselines.


**Outstanding**:

Despite the clarification regarding problem settings, the regression to a convergence rate of $\tilde{\mathcal{O}}(\epsilon^{-6})$ remains a fundamental objection for Reviewers rngr, AtLm, and kauD. The consensus is that such a slow rate limits the algorithmic impact and practical scalability, regardless of whether it is optimal for this specific fully stochastic formulation.

The attempt to address limited experimental breadth (Reviewers DoJP, AtLm) by adding WDRO-SI tasks was unconvincing. Closer inspection reveals that the Hessian-based baseline was improperly tuned (restricted stepsize to avoid instability), resulting in a flat line with zero convergence. Comparing against a paralyzed baseline does not constitute valid empirical evidence. Furthermore, the high variance observed in the proposed method validates concerns about the stability of the estimator.

Reviewer DoJP's suggestion to relax the strong convexity assumption to the PL condition remains unaddressed. The reliance on strong convexity significantly limits the scope of applicable problems compared to existing literature.


**Refs:**

[1] Lesi Chen, Jing Xu, and Jingzhao Zhang. On finding small hyper-gradients in bilevel optimization: Hardness results and improved analysis. CoLT, 2024.

[2]Maxime Bouscary, Jiawei Zhang, and Saurabh Amin. Reducing contextual stochastic bilevel optimization via structured function approximation. arXiv:2503.19991, 2025.

**Reviewer Scores:**

**Reviewer rngr (Score: 6 -> Est. unchanged):**

Reviewer rngr would likely keep the same overall score (6: marginally above acceptance). The authors provided a convincing argument regarding the per-iteration computational advantage of their first-order method, effectively answering the reviewer's question about competitiveness against second-order methods. This clarification reinforces the validity of the current positive rating by highlighting practical efficiency benefits that counterbalance the theoretical complexity gap noted by the reviewer. However, since the theoretical limitations (slower convergence rate in terms of $\epsilon$) are inherent to the method and remain unchanged, the score is unlikely to see a significant increase despite the solid rebuttal.

**Reviewer AtLm (Score: 4 -> Est. unchanged):**

Reviewer AtLm would likely keep the same overall score (4: marginally below acceptance). While the authors clarified the distinction regarding problem settings, the proposed $\tilde{\mathcal{O}}(\epsilon^{-6})$ rate still represents a significant theoretical regression compared to second-order baselines, which likely remains a major concern for the reviewer. Furthermore, although the authors addressed the request for broader experiments by adding WDRO-SI tasks, the results are unconvincing. The Hessian-based baseline in these new experiments exhibits almost zero convergence due to improper tuning (restricted stepsizes), while the proposed method demonstrates high variance. Consequently, the additional empirical evidence fails to robustly demonstrate the method's superiority or alleviate the reviewer's concerns regarding the efficiency gap.

**Reviewer DoJP (Score: 4 -> Est. unchanged):**

Reviewer DoJP would likely keep the same overall score (4: marginally below acceptance) or raise it marginally. The authors directly addressed the request for broader experimental validation by incorporating the Wasserstein Distributionally Robust Optimization (WDRO-SI) task and providing the requested comparisons with reduction-based methods. Furthermore, the authors clarified the necessity of the strong convexity assumption to avoid intractable three-level problems, responding to the query about PL conditions. Although the manuscript has been significantly strengthened, the reviewer's concern about the "open gap to optimal rate" due to variance bottlenecks remains a theoretical limitation that might prevent a strong jump in the score.

**Reviewer kauD (Score: 4 -> Est. unchanged):**

Reviewer kauD would likely keep the same overall score (4: marginally below acceptance). Similar to Reviewer AtLm, their primary critique stemmed from a comparison with complexity bounds from non-contextual or partially stochastic literature. While the authors have explained that their complexity matches the correct fully stochastic benchmarks and justified the double-loop structure as necessary for handling infinite contexts, such theoretical disagreements often persist unless the reviewer explicitly accepts the distinction in problem settings. Therefore, the evaluation will likely remain conservative despite the clarifications.

---

### Decision · Program_Chairs · 2026-01-26

Reject